# Continuous home monitoring of Parkinson's disease using inertial sensors: A systematic review

Marco Sica[1]*, Salvatore Tedesco[1], Colum Crowe[1], Lorna Kenny[2], Kevin Moore[2], Suzanne Timmons[2], John Barton[1], Brendan O'Flynn[1], Dimitrios-Sokratis Komaris[1]

1 Tyndall National Institute, University College Cork, Cork, Ireland, 2 Centre for Gerontology and Rehabilitation, University College Cork, Cork, Ireland

* marco.sica@tyndall.ie

**Data Availability Statement:** All relevant data are within the manuscript and its Supporting Information files.

## Abstract

Parkinson's disease (PD) is a progressive neurological disorder of the central nervous system that deteriorates motor functions, while it is also accompanied by a large diversity of non-motor symptoms such as cognitive impairment and mood changes, hallucinations, and sleep disturbance. Parkinsonism is evaluated during clinical examinations and appropriate medical treatments are directed towards alleviating symptoms. Tri-axial accelerometers, gyroscopes, and magnetometers could be adopted to support clinicians in the decision-making process by objectively quantifying the patient's condition. In this context, at-home data collections aim to capture motor function during daily living and unobstructedly assess the patients' status and the disease's symptoms for prolonged time periods. This review aims to collate existing literature on PD monitoring using inertial sensors while it focuses on papers with at least one free-living data capture unsupervised either directly or via video-tapes. Twenty-four papers were selected at the end of the process: fourteen investigated gait impairments, eight of which focused on walking, three on turning, two on falls, and one on physical activity; ten articles on the other hand examined symptoms, including bradykinesia, tremor, dyskinesia, and motor state fluctuations in the on/off phenomenon. In summary, inertial sensors are capable of gathering data over a long period of time and have the potential to facilitate the monitoring of people with Parkinson's, providing relevant information about their motor status. Concerning gait impairments, kinematic parameters (such as duration of gait cycle, step length, and velocity) were typically used to discern PD from healthy subjects, whereas for symptoms' assessment, researchers were capable of achieving accuracies of over 90% in a free-living environment. Further investigations should be focused on the development of ad-hoc hardware and software capable of providing real-time feedback to clinicians and patients. In addition, features such as the wearability of the system and user comfort, set-up process, and instructions for use, need to be strongly considered in the development of wearable sensors for PD monitoring.

**Funding:** This manuscript has emanated from research supported by the European Regional Development Fund (ERDF) under Ireland's European Structural and Investment Funds Programme 2014-2020. Aspects of this work have been supported in part by INTERREG NPA funded project SenDOC. Aspects of this publication were supported by Enterprise Ireland and Abbvie Inc. under grant agreement no. IP 2017 0625. The funders had no role in study design, data collection and analysis, decision to publish, or preparation of the manuscript.

**Competing interests:** The authors have read the journal's policy and have the following competing interests: Abbvie Inc. provided partial funding for this study. This does not alter our adherence to PLOS ONE policies on sharing data and materials. There are no patents, products in development or marketed products associated with this research to declare.

**Abbreviations:** ANN, Artificial Neural Networks; DD, Diverse Density; EM-DD, Expectation Maximization-Diverse Density; FoG, Freezing-of-gait; ID-APR, Discriminative variant of the axis-parallel hyper-rectangle; IMU, Inertial Measurement Unit; m-AIMS, modified-Abnormal Involuntary Movements; MDS-UPDRS, Movement Disorder Society-Unified Parkinson's Disease Rating Scale; MIL-kNN, Multiple instance learning k-Nearest Neighbor; MI-SVM, Multiple Instance Support Vector Machine; MVPA, moderate to vigorous physical activities; PD, Parkinson's disease; PASE, Physical Activity Scale in the Elderly; PwP, people with Parkinson's; SVM, Support Vector Machine.

## Introduction

Parkinson's disease (PD) is a chronic neurological disorder of the central nervous system. Its incidence rises dramatically with age, affecting approximately 6.2 million people worldwide in 2015 [1]. The symptoms of PD are multiple, with the most identifiable being related to motor degeneration. In general, they appear gradually and become more evident with the worsening of the disease, varying from person to person. The diagnosis of PD can be challenging, especially at an early stage, due to the lack of specific tests [2]. The most recognizable symptoms include tremor, rigidity, bradykinesia, and postural instability [3].

Tremor typically appears at the distal part of the limbs, affecting a single arm or leg; it is more pronounced in the upper extremities and it progresses bilaterally with the degeneration of the disease. Rigidity refers to an immoderate, continuous contraction of muscles, and an increased resistance to joint movements. Bradykinesia, as a general term, can be differentiated into akinesia, bradykinesia and hypokinesia, indicating absence, slow or decreased bodily movements, respectively. Akinesia may also include the freezing-of-gait (FoG) phenomenon, which causes sudden and temporary episodes of inability to move forward despite the intention to walk. Postural instability is related to loss of balance and the inability to maintain the upright position, often causing falls or a fear of falling [3].

Despite PD being an irreversible neurodegenerative disorder, medications, such as Levodopa can provide symptomatic relief, particularly in earlier stages [4]. The "on" and "off" phenomenon in Levodopa-treated patients, describes motor fluctuations that occur as the levels of dopamine in the brain drop, followed by a worsening of the motor function: during the "on" state the symptoms are well managed, while in the "off" state they deteriorate. In newly diagnosed people with Parkinson's (PwP), the response to a single drug intake may last for several hours, whereas with the progression of the disease the drug's effect is shortened (4 hours or less), and patients need to decrease intervals between doses and/or increase the dosages [5, 6]. Drug-induced dyskinesia (i.e. involuntary abnormal muscle movements [7]) can appear during the "on" state in some patients who have been taking Levodopa for a prolonged period of time.

To ensure the appropriate medical treatment and correct dose of medication for an individual, PwP are infrequently evaluated with qualitative clinical assessments that are based on the subjective judgment of specialists, such as the Movement Disorder Society—Unified Parkinson's Disease Rating Scale (MDS-UPDRS), or specifically for dyskinesia, the modified Abnormal Involuntary Movement Scale (m-AIMS) [8, 9]. Yet, due to the heterogeneity and complexity of PD symptoms, such clinical assessments can be challenging and time consuming. Clinicians with different backgrounds and experiences might also vary in their interpretations of the MDS-UPDRS and m-AIMS [10]. Equally, a person's motor state at a clinic appointment may not be typical of their usual state, enhanced by fatigue, dehydration from travelling or anxiety [10]. Therefore, a clinical assessment is only a snapshot in time, giving little indication of function in a more on or off state. Ultimately, the only way to properly characterize a patient's motor status is to continuously evaluate their motor function over an extended period of time.

Due to their small-size, light weight, and low-power, wearable motion sensors have already demonstrated their clinical relevance in healthcare [11–13] and daily-life monitoring [14, 15]. The most widely used sensors are tri-axial accelerometers, gyroscopes, and magnetometers, commonly combined in an inertial measurement unit (IMU) that can capture three-dimensional orientation, and linear and angular velocities [16, 17]. Thanks to the development of miniaturized hardware technologies capable of collecting and storing large amount of raw data [18], IMUs may offer the opportunity to improve the evaluation of the PD motor symptoms

by collecting free-living movements for prolonged period of time outside the laboratory environment. Former studies, such as the one by Bloem et al. [19], have reported that PwP walk better when observed rather than when unsupervised in their daily lives. This is a consequence of the well-known "Hawthorne observation effect" [20]: free-living activities involve a combination of tasks with varying complexities, challenges and distractions that may reduce attention. In addition, numerous episodes related with PD are challenging to detect during laboratory-based observation because of their complexity (i.e. the on/off phenomenon) or rarity (i.e. freezing of gait phenomenon) [21]. As a consequence, a thorough evaluation of a PwP requires the data to be gathered during long observation windows while patients go ahead with normal every day activities.

Previous reviews have already investigated monitoring of PD using body-fixed-sensors [22–28]; yet, to our best knowledge, this is the first systematic review to target solely publications on continuous monitoring of PwP with at least one data capture at home. We focused on studies that used only wearable inertial sensor over a long period of time (i.e. from one to fourteen days) and where the data collection was not supervised (either directly or via videotape) by clinicians or caregivers.

## Methodology

This systematic review was performed according to the guidelines of the PRISMA statement [29]. The literature search was conducted in April 2020 on the IEEE Xplore, PubMed, SpringerLink, ACM Digital Library and Web of Science electronic databases with the following search string:

> (Parkins*) AND (bradykinesia OR tremor OR rigidity OR hypokinesia OR dyskinesia OR freez* OR akinesia OR fluctuat* OR movement disorder) AND (IMU or inertia* OR acceler* OR gyro* OR wearable OR body-worn) AND (free-living OR daily-living OR continuous OR 24-hour OR home OR unsupervised)

Only original, full-text, peer-reviewed, journal or conference articles in English that were published between January 2010 and April 2020 were included in this review. Case studies, reviews, books, book chapters, editorials, and letters were excluded. Duplicate findings were manually identified and removed.

Three reviewers (MS, ST, and CC) independently screened the title, abstract and key words of the records identified through the database searching. Studies were selected if they monitored or estimated the severity of PD symptoms at home with inertial sensors and their data collection was not supervised by research staff or video cameras. Studies were excluded if the main recording devices were not IMUs, or PD was not the prevalent disorder of the sample population. Subsequently, full text assessment was performed by each reviewer and cases of conflict were debated among them.

The relevant data was extracted from chosen studies and tabularized under predefined headings. Authorship, symptoms monitored, activities, devices (type, number, placement) and data collection (number of assessment days, sample size, use of diaries) were all recorded. Additionally, the studies' aims, outcome measures, analyses used and results were summarized.

To analyze the risk of bias of the reviewed studies, an adapted version of the AXIS appraisal tool for cross-sectional studies was used, containing thirteen questions that could be answered with a "yes" or "no" [30] (Table 1). A single reviewer scored each study from zero to 13 against the appraisal tool by summing all the positive answers. Papers were categorized as having low

**Table 1. Risk and quality assessment questions.**

| Question number | AXIS question code | |
|---|---|---|
| | | **INTRODUCTION** |
| Q1 | 1 | Were the aims/objectives of the study clear? |
| | | **METHODS** |
| Q2 | 2 | Was the study design appropriate for the stated aim(s)? |
| Q3 | 3, 4 & 5 | Was the sample size justified, clearly defined, and taken from an appropriate population? |
| Q4 | 6 | Was the selection process likely to select subjects/participants that were representative of the target/reference population under investigation? |
| Q5 | 8 | Were the outcome variables measured appropriate to the aims of the study? |
| Q6 | 9 | Were the outcome variables measured correctly using instruments/measurements that had been trialed, piloted or published previously? |
| Q7 | 10 | Is it clear what was used to determined statistical significance and/or precision estimates? (e.g. p-values, confidence intervals) |
| Q8 | 11 | Were the methods (including statistical methods) sufficiently described to enable them to be repeated? |
| | | **RESULTS** |
| Q9 | 12 | Were the basic data adequately described? |
| Q10 | 16 | Were the results presented for all the analyses described and presented in the methods? |
| | | **DISCUSSION** |
| Q11 | 17 | Were the authors' discussions and conclusions justified by the results? |
| Q12 | 18 | Were the limitations of the study discussed? |
| | | **OTHER** |
| Q13 | 19 | Were there any funding sources or conflicts of interest that may affect the authors' interpretation of the results? |

(score equal or higher than 11), medium (score between eight and 10) and high (score equal or lower than seven) risk of bias.

## Results

### Studies selection

The electronic database searches identified 446 records (Fig 1). Ninety-eight duplicates were removed and the remaining 348 articles were screened (229 records excluded). Following full text assessment (95 records excluded) a total of 24 studies were included in the review [31–54].

### Risk of bias assessment

The appraisal tool yielded six studies with medium and 18 with low risk of bias. Authors reported clear aims and objectives (Q1, 95.8%), study designs (Q2, 95.8%) and selection processes (Q4, 83.3%), however, the sample size was inadequate in 37.5% of the cases (Q3). The outcome variables were appropriate to the aims (Q5, 100%) and measured with the correct instruments (Q6, 100%), while statistics and general methods were reported adequately (Q7, 87.5%; Q8, 79.1%). Results were presented in depth (Q9, 87.5%) and described in the methods (Q10, 87.5%). Discussions and conclusions were justified by the results (Q11, 100%) with no conflicts of interests (Q13, 100%), yet, 37.5% of the authors omitted or did not fully investigate the study's limitations (Q12). Detailed scores for each level of bias and each individual study are presented in S1 and S2 Tables.

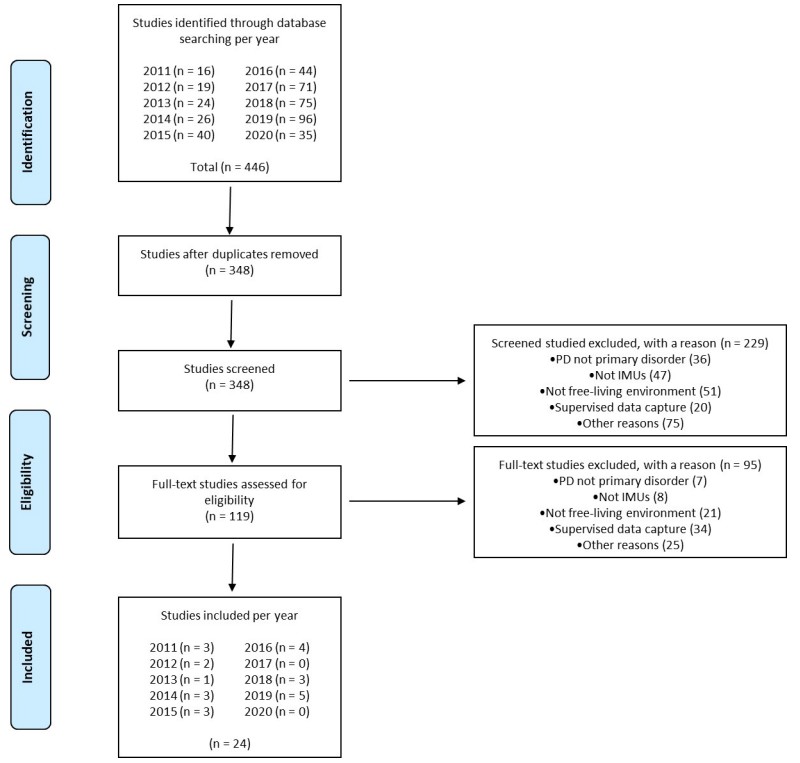

**Fig 1. PRISMA flow diagram.**

## Characteristics of included studies

Among the 24 identified papers, 15 included sessions that were recorded both at home and in the lab during one [33, 35, 38–45, 51], two [31, 32], or three visits [36, 37]; these studies included an initial calibration/validation in a supervised environment for the development of ad-hoc algorithms and then subsequent implementation/testing in an unsupervised setting. Besides the number of testing days and lab sessions, the exact duration of each lab assessment was not always specified, thus hindering the reproducibility of the protocol. In contrast, nine articles described data collection that was exclusively undertaken at home [34, 46–50, 52–54] (Table 2).

Sample sizes ranged from seven [31, 41] to 125 [51] PwP, and from nine [31] to 67 [39] controls in laboratory environments. The same numbers for at-home tests ranged from one [33, 52] to 170 [50] PwP, and from one [33] to 172 [50] controls. In 13 cases, PwP were asked to fill a diary in order to track activities[31, 36, 37, 53], medication intake [31, 32, 34, 35] and symptoms [34, 41, 45–47, 52, 54] (Table 2).

Data were collected by accelerometers alone (in 46.7% of the studies recording in labs and 50% of the studies at-home) [33, 35–37, 41, 44–47, 49–51, 53], in combination with gyroscopes (lab: 33.3%, home: 33.3%) [31–33, 39, 42, 43, 48, 52] or along with gyroscopes and magnetometers (lab: 20%, home: 16.7%) [34, 38, 40, 51, 54]. Authors used off-the-shelf devices such as the AX3 (Axivity, York, UK) [44, 45, 50, 51], DynaPort (McRoberts, The Hague, Netherlands) [33, 39, 42, 43, 48], GT3X (*ActiGraph*, Pensacola, USA) [53], Mimamori-gait system (Mitsubishi Chemical, Tokyo, Japan) [36, 37], Mobi8 (TMSI International, Oldenzaal, The Netherlands) [33], MOX5 (Maastricht Instruments, Maastricht, The Netherlands) [52], Opal (APDM, Portland, USA) [34, 38, 40, 51, 54] and Parkinson's Kinetigraph (Global Kinetics

**Table 2. Characteristics of the studies.**

| Author/Study | | Device | Sampling frequency | N° of Sensors | Accelerometer | Gyroscope | Magnetometer | Sensors Position | Assessment | Patients | Controls | Diary Activities | Diary Medication | Diary Symptoms | Bradykinesia | Tremor | On/Off State | Dyskinesia | Walking | Turning | Fall Detection | Physical Activity |
|---|---|---|---|---|---|---|---|---|---|---|---|---|---|---|---|---|---|---|---|---|---|---|
| Moore et al. (2011) [31] | Controlled environment | Prototype | 100 Hz | 1 | ✓ | ✓ | - | Left shank | 2 lab sessions | 4 / 3 | 9 / - | - | - | - | | | | | ✓ | | | |
| | Home | Prototype | 100 Hz | 1 | ✓ | ✓ | - | Left shank | 1 day | 3 | - | ✓ | ✓ | - | ✓ | | | | | | | |
| Pastorino et al. (2011) [32] | Controlled environment | Prototype | 62.5 Hz | 5 | ✓ | ✓ | - | Both wrists both ankles waist | 2 lab sessions | 20 / 12 | 10 / - | - | - | - | | | | | | | | |
| | Home | Prototype | 62.5 Hz | 5 | ✓ | ✓ | - | Both wrists both ankles waist | 7 days | 24 | - | - | ✓ | - | | | | | | | | |
| Weiss et al. (2011) [33] | Controlled environment | Mobi8 | 256 Hz | 1 | ✓ | - | - | Lower back | 1 lab session | 22 | 17 | - | - | - | | | | | ✓ | | | |
| | Home | DynaPort | N/A | 1 | ✓ | ✓ | - | Lower back | 3 days | 1 | 1 | - | - | - | | | | | | | | |
| Das et al. (2012) [34] | Home | Opal | 40 Hz | 5 | ✓ | ✓ | ✓ | Both wrists both ankles waist | 4 days | 2 | - | - | ✓ | ✓ | | ✓ | | ✓ | | | | |
| Griffiths et al. (2012) [35] | Controlled environment | Parkinson's Kinetigraph | 50 Hz | 1 | ✓ | - | - | Wrist (most affected side) | 1 lab session | 34 | 10 | - | - | - | ✓ | | | ✓ | | | | |
| | Home | Parkinson's Kinetigraph | 50 Hz | 1 | ✓ | - | - | Wrist (most affected side) | 10 days | 34 | 10 | - | ✓ | - | | | | | | | | |

| Author/Study | | Device | Sampling frequency | N° of Sensors | Accelerometer | Gyroscope | Magnetometer | Sensors Position | Assessment | Patients | Controls | Diary Activities | Diary Medication | Diary Symptoms | Bradykinesia | Tremor | On/Off State | Dyskinesia | Walking | Turning | Fall Detection | Physical Activity |
|---|---|---|---|---|---|---|---|---|---|---|---|---|---|---|---|---|---|---|---|---|---|---|
| Yoneyama et al. (2013/2014)(Part 1 and 2) [36, 37] | Controlled environment | Mimamori-gait system | 100 Hz | 1 | ✓ | - | - | Waist (front centre) | 3 lab sessions | - / - / 12 | 11 / 1 / - | - | - | - | | | | | | | | |
| | Home | Mimamori-gait system | 100 Hz | 1 | ✓ | - | - | Waist (front centre) | 1 day | 17 | 10 | ✓ | - | - | | | | | ✓ | | | |
| El-Gohary et al. (2014) [38] | Controlled environment | Opal | 128 Hz | 1 | ✓ | ✓ | ✓ | Lower back | 1 lab session | 21 | 19 | - | - | - | | | | | ✓ | ✓ | | |
| | Home | Opal | 128 Hz | 3 | ✓ | ✓ | ✓ | Lower back On top of each foot | 7 days | 12 | 18 | - | - | - | | | | | | | | |
| Weiss et al. (2014) [39] | Controlled environment | DynaPort | 100 Hz | 1 | ✓ | ✓ | - | Lower back | 1 lab session | 40 PD faller | 67 PD Non faller | - | - | - | | | | | | | ✓ | |
| | Home | DynaPort | 100 Hz | 1 | ✓ | ✓ | - | Lower back | 3 days | 40 PD faller | 67 PD Non faller | - | - | - | | | | | ✓ | ✓ | | |
| Mancini et al. (2015) [40] | Controlled environment | Opal | N/A | 3 | ✓ | ✓ | ✓ | Lower back On top of each foot | 1 lab session | 13 | 19 | - | - | - | | | | | ✓ | | | |
| | Home | Opal | N/A | 3 | ✓ | ✓ | ✓ | Lower back On top of each foot | 7 days | 13 | 19 | - | - | - | | | | | ✓ | ✓ | | |

(Continued)

**Table 2.** (Continued)

| Author/Study | Environment | Device | Sampling frequency | N° of Sensors | Accelerometer | Gyroscope | Magnetometer | Sensors Position | Assessment | Patients | Controls | Diary: Activities | Diary: Medication | Diary: Symptoms | Bradykinesia | Tremor | On/Off State | Dyskinesia | Walking | Turning | Fall Detection | Physical Activity |
|---|---|---|---|---|---|---|---|---|---|---|---|---|---|---|---|---|---|---|---|---|---|---|
| Pérez-López et al. (2015) [41] | Controlled environment | Prototype | 40 Hz | 1 | ✓ | - | - | Waist (left lateral side) | 1 lab session | 7 | - | - | - | - | | | ✓ | | | | | |
| | Home | Prototype | 40 Hz | 1 | ✓ | - | - | Waist (left lateral side) | 1 days | 7 | - | - | - | ✓ | | | | | | | | |
| Weiss et al. (2015) [42] | Controlled environment | DynaPort | 100 Hz | 1 | ✓ | ✓ | - | Lower back | 1 lab session | 28 PD Freezers | 44 PD Non-Freezers | - | - | - | | | | | ✓ | | | |
| | Home | DynaPort | 100 Hz | 1 | ✓ | ✓ | - | Lower back | 3 days | 28 PD Freezers | 44 PD Non-Freezers | - | - | - | | | | | | | | |
| Bernad-Elazari et al. (2016) [43] | Controlled environment | DynaPort | 100 Hz | 1 | ✓ | ✓ | - | Lower back | 1 lab session | 99 | 38 | - | - | - | | | | | ✓ | | | |
| | Home | DynaPort | 100 Hz | 1 | ✓ | ✓ | - | Lower back | 3 days | 99 | 38 | - | - | - | | | | | | | | |
| Del Din et al. (2016) [44] | Controlled environment | AX3 | 100 Hz | 1 | ✓ | - | - | Lower back | 1 lab session | 47 | 50 | - | - | - | | | | | ✓ | | | |
| | Home | AX3 | 100 Hz | 1 | ✓ | - | - | Lower back | 7 days | 47 | 50 | - | - | - | | | | | | | | |
| Fisher et al. (2016) [45] | Controlled environment | AX3 | 100 Hz | 2 | ✓ | - | - | Both wrists | 1 lab session | 34 | - | - | - | - | | | ✓ | ✓ | | | | |
| | Home | AX3 | 100 Hz | 2 | ✓ | - | - | Both wrists | 7 days | 34 | - | - | - | ✓ | | | | | | | | |
| Ossig et al. (2016) [46] | Home | Parkinson's Kinetigraph | 50 Hz | 1 | ✓ | - | - | Wrist (most affected side) | 1 day | 24 | - | - | - | ✓ | | | ✓ | ✓ | | | | |
| Battista and Romaniello et al. (2018) [47] | Home | Prototype | 100 Hz | 1 | ✓ | - | - | Wrist (most affected side) | 1 day | 3 | - | - | - | ✓ | | ✓ | | | | | | |
| Mancini et al. (2018) [48] | Home | DynaPort | 100 Hz | 1 | ✓ | ✓ | - | Lower back | 3 days | 69 PD Freezers | 25 PD Non-Freezers | - | - | - | | | | | ✓ | ✓ | | |
| Rodriguez-Molinero et al. (2018) [49] | Home | Prototype | 40 Hz | 1 | ✓ | - | - | Waist (left lateral side) | From 1 to 3 days | 23 | - | - | - | ✓ | | | ✓ | | | | | |
| Del Din et al. (2019) [50] Del Din et al. (2019) [50] | Home | AX3 | 100 Hz | 1 | ✓ | - | - | Lower back | 7 days | 170 PD faller | 172 Non faller | - | - | - | | | | | ✓ | | ✓ | |
| Galperin et al. (2019) [51] | Controlled environment | Opal | N/A | 1 | ✓ | ✓ | ✓ | Lower back | 1 lab session | 125 | - | - | - | - | | | | | ✓ | | | |
| | Home | AX3 | 100 Hz | 1 | ✓ | - | - | Lower back | 7 days | 125 | - | - | - | - | | | | | | | | |
| Heijmans et al. (2019) [52] | Home | MOX5 | 200 Hz | 2 | ✓ | ✓ | - | Both wrists | 14 days | 1 | - | - | - | ✓ | | ✓ | | | | | | |
| Mantri et al. (2019) [53] | Home | Actigraph GT3X | N/A | 1 | ✓ | - | - | Waist | 7 days | 29 | - | ✓ | - | - | | | | | | | | ✓ |

*(Continued)*

**Table 2.** (Continued)

| Author/Study | | Device | Sampling frequency | N° of Sensors | Sensors | | | Sensors Position | Assessment | Subjects | | Diary | | | Symptoms and Fluctuations | | | | Gait Impairments | | | Physical Activity |
|---|---|---|---|---|---|---|---|---|---|---|---|---|---|---|---|---|---|---|---|---|---|---|
| | | | | | Accelerometer | Gyroscope | Magnetometer | | | Patients | Controls | Activities | Medication | Symptoms | Bradykinesia | Tremor | On/ Off State | Dyskinesia | Walking | Turning | Fall Detection | |
| McNames et al. (2019) [54] | Home | Opal | 128 Hz | 2 | ✓ | ✓ | ✓ | Both wrists | 7 days | 10 | 7 | - | - | - | | ✓ | | | | | | |

Abbreviations: lab = laboratory, PD = Parkinson's disease.

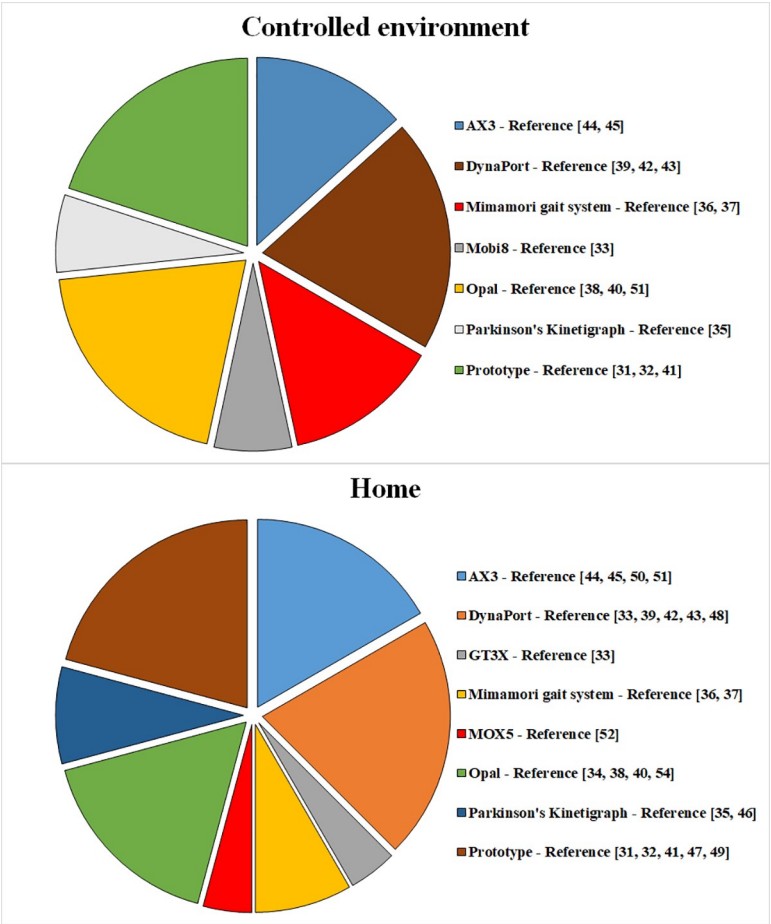

**Fig 2. Device–Number of studies.**

Corporation, Melbourne, Australia) [35, 36]. In five studies, volunteers wore prototype sensors [31, 32, 41, 47, 49] (Fig 2). Data collection frequently lasted for a week and ranged from one [31, 36, 37, 41, 46, 47, 49] to 14 days [52] (Fig 3 and Table 2).

Fourteen works investigated gait impairments, eight of which focused on walking [31, 33, 36, 37, 42–44, 51], three on turning [38, 40, 48], two on falls [39, 50] and one on physical

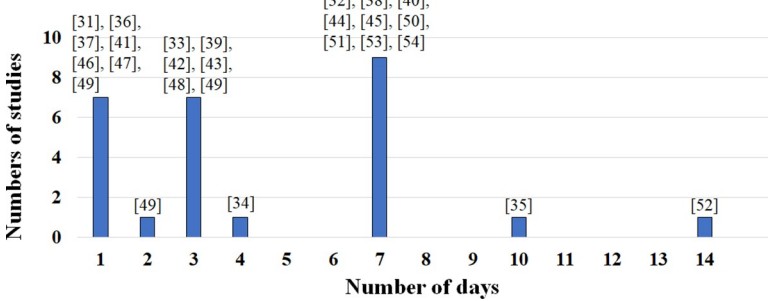

**Fig 3. Data collection–Number of days.**

**Table 3. Aim, outcome measures, type of analyses, and results of the studies.**

| Author Study | Aim | Outcome measures | Analysis | Results | |
|---|---|---|---|---|---|
| **Moore et al. (2011) [31]** | PD gait analysis | Stride length (m) | Walk detection algorithm<br><br>2-D Plots (24 h activities) | **Controlled environment** | Mean stride length error was equal to $0.064 \pm 0.013$ m for controls and $0.045 \pm 0.024$ m for PD patients. Moreover, there were considerable fluctuations in stride length for patients with a longer duration of the disease |
| | | | | **Home** | Fluctuation in stride length (as in controlled environment) |
| **Pastorino et al. (2011) [32]** | Detection of bradykinesia severity | Bradykinesia episodes (starting time and duration)<br><br>Bradykinesia Score [0–4] | Activity recognition algorithm<br>Bradykinesia estimation: SVM classifier and meta-analysis algorithm | **Controlled environment** | - |
| | | | | **Home** | The accuracy between the Bradykinesia Score outcome and the UPDRS (clinicians twice per day) was equal to $68.3 \pm 8.9\%$ with the standard classifier and $74.4 \pm 14.9\%$ with meta-analysis algorithm. |
| **Weiss et al. (2011) [33]** | PD gait analysis | Temporal measures: average stride time (s)<br><br>Frequency measures: stride time variability (%), dominant frequency (Hz), amplitude (psd), width (Hz), and slope (psd/Hz) | 3-D Plots of the amplitude in the frequency domain | **Controlled environment** | Less consistent walking patterns in PD patients compared to controls. Moreover, the frequency amplitude was smaller in PD patients ($0.67 \pm 0.22$ psd) than in controls ($0.94 \pm 0.16$ psd) |
| | | | | **Home** | Results extended in an unsupervised environment. In particular, frequency amplitude above 0.3 psd for only 165 minutes in PD and frequency amplitude above 0.3 psd for 355 minutes in controls. |
| **Das et al. (2012) [34]** | Detection of tremor and dyskinesia episodes | Tremor episodes (starting time and duration) Dyskinesia episodes (starting time and duration) | Dyskinesia and tremor estimation: ID-APR, MI-SVM, kNN, DD, and EM-DD classifiers | **Home** | ID-APR classifier achieved the best performances with an accuracy (outcome vs symptoms diary) over 90% for both dyskinesia and tremor. |
| **Griffiths et al. (2012) [35]** | Detection of bradykinesia and dyskinesia severity | Bradykinesia episodes (starting time and duration)<br><br>Dyskinesia episodes (starting time and duration)<br><br>Bradykinesia Score [0–80]<br>Dyskinesia Score [0–80] | Bradykinesia and dyskinesia estimation: fuzzy logic algorithms. | **Controlled environment** | The Bradykinesia Score outcome (compared to the Bradykinesia Score dot test) had a specificity of 88% and a sensitivity of 95%.<br><br>The Dyskinesia Score outcome had a highly significant correlation with the AIMS test (Pearson's $p < 0.0001$, R of 0.80). |
| | | | | **Home** | Correlation ($p < 0.05$) between global median bradykinesia (from 10 days recording) and UPDRS.<br><br>Correlation ($p < 0.0005$) between global median dyskinesia (from 10 days recording) and UPDRS |
| **Author Study** | **Aim** | **Outcome measures** | **Analysis** | **Results** | |
| **Yoneyama et al. (2013) (Part 1 and 2) [36, 37]** | PD gait analysis | Number gait peaks, gait cycle (s) and average acceleration magnitude per cycle (m/s$^2$) | Walk detection algorithm | **Controlled environment** | The accuracy of the gait peaks detection between the outcome and the videotape was over 94% |
| | | | | **Home** | Average gait cycle was larger in PD ($1.16 \pm 0.20$ s) rather than controls ($1.08 \pm 0.19$ s). In addition, the recognition of PD gait from a normal gait had 100% sensitivity, 94.1% specificity, and 96.3% accuracy. |

*(Continued)*

**Table 3.** (Continued)

| Author Study | Aim | Outcome measures | Analysis | Results | |
|---|---|---|---|---|---|
| **El-Gohary et al. (2014)** [38] | PD turning analysis | Number of bouts/h, duration (s), step duration (s), active-rate (%), number of turns, number of turns/h, duration (s) angle (degrees), peak velocity (degrees/s), and number of steps | Turn detection algorithm | **Controlled environment** | The turn detection algorithm achieved a sensitivity of 90% and 76% and a specificity of 75% and 65% when compared respectively with a motion analysis system and a videotape. |
| | | | | **Home** | PD tend to take shorter turns with smaller turn angles and more steps than controls. |
| **Weiss et al. (2014)** [39] | PD fall risk and gait analysis | Total number of activity bouts, total percent of activity duration (%), total number of steps for 3-days, median activity bout duration (s), median number of steps for bout, cadence (steps/min), amplitude of dominant frequency (prs), width of dominant frequency (Hz), stride regularity ($g^2$), and harmonic ratio | Walk detection algorithm | **Controlled environment** | - |
| | | | | **Home** | The walking quantity is similar between PD fallers and non-fallers, while fallers had a higher step to step variability. |
| | | | | | Outcomes measures predicted the time to first fall (p = 0.0034) in PD patients who reported no falls in the year prior to testing. |
| **Mancini et al. (2015)** [40] | PD turning analysis | Active rate (%),number of turns, number of turns/hour, turn angle (degrees), CV turn angle, turn duration (s), CV turn duration, number of steps /turn, CV number of steps /turn, turn mean velocity (degrees/s), and CV turn mean velocity | Turn detection algorithm | **Controlled environment** | Velocity and turn detection were similar (outcomes vs observed events) in healthy and PD subjects (p = 0.34 and p = 0.33) |
| | | | | **Home** | PD patients realized the turning movement slower than the controls (turn mean velocity 38 ± 5.7°/s and 43.3 ± 4.8°/s, respectively) with a major number of steps (mean number of steps 3.2 ± 0.8 and 1.7 ± 1.1, respectively) |
| **Pérez-López et al. (2015)** [41] | Detection of ON/OFF state | ON/OFF episodes (starting time and duration) | Walk detection algorithm | **Controlled environment** | - |
| | | Bradykinesia (starting time and duration) Dyskinesia (starting time and duration) | Bradykinesia and dyskinesia estimation: using thresholds (frequency analysis). | **Home** | ON/OFF classifier, compared to the self-recorded motor state, had a sensitivity of 99.9% and a specificity of 99.9% |
| | | | ON state detection: when dyskinesia is detected. | | |
| | | | OFF state detection: when bradykinesia is detected. | | |
| **Weiss et al. (2015)** [42] | PD gait analysis in patients suffering of freezing of gait and not | Total number of activity bouts, total percent of activity duration (%), total number of steps for 3-days, median activity bout duration (s), median number of steps for bout, and cadence (steps/min), amplitude of dominant frequency (prs), width of dominant frequency (Hz), stride regularity ($g^2$), and harmonic ratio | Walk detection algorithm | **Controlled environment** | - |
| | | | | **Home** | Freezers' walkers had a higher gait variability (i.e., the anterior–posterior power spectral density width; p = 0.003) and a lower gait consistency (i.e., the vertical stride regularity; p = 0.007) |
| Author Study | Aim | Outcome measures | Analysis | Results | |

(*Continued*)

**Table 3.** (*Continued*)

| Author Study | Aim | Outcome measures | Analysis | | Results |
|---|---|---|---|---|---|
| **Bernad-Elazari et al. (2016)** [43] | Assessment of PD conditions | Classification PD vs Healthy, PD mild vs PD severe, and PD mild vs Healthy | Recognition of walk-to-sit and sit-to-walk transitions. | **Controlled environment** | PD vs Healthy: accuracy = 74.6% |
| | | | | | PD mild vs PD severe: accuracy = 56.2% |
| | | | SVM to discriminate different PD conditions | | PD mild vs Healthy: accuracy = 52.7 |
| | | | | **Home** | PD vs Healthy: accuracy = 92.3% |
| | | | | | PD mild vs PD severe: accuracy = 89.8% |
| | | | Leave-one-out approach | | PD mild vs Healthy: accuracy = 85.9% |
| **Del Din et al. (2016)** [44] | PD gait analysis | Step velocity (m/s), step length (m), swing time var (s), step velocity var (m/s), step length var (m), step time var (s), stance time var (s), step time (s), swing time (s), stance time (s), step time asy (s), swing time asy (s), stance time asy (s), and step length asy (m) | Walk detection algorithm | **Controlled environment** | 2 out of 14 outcomes were significantly different in PD and controls. |
| | | | | | PD patients walked with slower and shorter steps (i.e., step velocity 1.254 ± 0.211 m/s and 1.393 ± 0.207 m/s for PD and controls, respectively) |
| | | | | **Home** | 4 out of 14 outcomes were significantly different in PD and controls. |
| | | | | | PD patients walked with slower and shorter steps (i.e., step velocity 1.038 ± 0.422 m/s and 1.103 ± 0.411 m/s for PD and controls, respectively) |
| **Fisher et al. (2016)** [45] | Detection of ON/OFF state and Dyskinesia episodes | ON/OFF episodes (starting time and duration) | ON/OFF state and dyskinesia estimation: ANN and leave-one-out approach | **Controlled environment** | Classification algorithm vs diary: |
| | | | | | ON: sensitivity = 69%, specificity = 82% |
| | | | | | OFF: sensitivity = 60%, specificity = 83% |
| | | Dyskinesia episodes (starting time and duration) | | | Dyskinesia: sensitivity = 49%, specificity = 99% |
| | | | | **Home** | Diary vs ANN |
| | | | | | ON: sensitivity = 52%, specificity = 91% |
| | | | | | OFF: sensitivity = 50%, specificity = 83% |
| | | | | | Dyskinesia: sensitivity = 38%, specificity = 93% |
| **Ossig et al. (2016)** [46] | Detection of ON/OFF state and Dyskinesia episodes | Dyskinesia episodes (starting time and duration) | ON/OFF state and dyskinesia estimation via calibrated individual thresholds | **Home** | The classifier ON/OFF and Dyskinesia, compared to the diary, had a moderate-to-strong correlation (p from 0.404 to 0.658) |
| | | Bradykinesia episodes (starting time and duration) | | | |
| | | ON/OFF episodes (starting time and duration) | | | |
| **Battista and Romaniello et al. (2018)** [47] | Detection of tremor episodes | Tremor episodes (starting time and duration) | Tremor detection: using thresholds (frequency analysis) | **Home** | Tremor outcome, compared to the diary, had a sensitivity of 99.3%, a specificity of 99.6%, and an accuracy of 98.9% |
| **Mancini et al. (2018)** [48] | PD turning analysis in patients suffering of freezing of gait and not | Number of turns/30 min, turn angle (degrees), CV turn angle, turn duration (s), CV turn duration, mean velocity (degrees/s), CV mean velocity, peak velocity (degrees/s), CV peak velocity, 2D jerk (m$^2$/s$^5$), CV 2D jerk, ML jerk (m$^2$/s$^5$), CV ML jerk, ML range (m$^2$/s), and CV ML range | Turn detection algorithm | **Home** | Similar number of turns in PD freezers and non-freezers: 19.3 ± 9.2 /30 min and 22.4 ± 12.9 /30 min respectively (p = 0.194). Furthermore, mean jerkiness, mean and variability of medio-lateral jerkiness were higher in freezers (p < 0.05). |
| **Author Study** | **Aim** | **Outcome measures** | **Analysis** | | **Results** |

(*Continued*)

**Table 3.** (Continued)

| Author Study | Aim | Outcome measures | Analysis | | Results |
|---|---|---|---|---|---|
| **Rodriguez-Molinero et al. (2018) [49]** | Detection of ON/OFF state | ON/OFF episodes (starting time and duration) | Walk detection algorithm Bradykinesia and dyskinesia detection: using thresholds (frequency analysis). | **Home** | The accuracy between the classifier algorithm and the diary was equal to 92.20% |
| | | Bradykinesia (starting time and duration) | ON state detection: when dyskinesia is detected. | | |
| | | Dyskinesia (starting time and duration) | OFF state detection: when bradykinesia is detected. | | |
| **Del Din et al. (2019) [50]** | PD fall risk and gait analysis | Macro gait: total walking, time per day (min), percentage of walking time, number of steps per day, Bouts per day, mean bout length (sec), and variability (S2 Table). | Walk detection algorithm | **Home** | PD fallers had a greater variability (step length) while controls fallers less variability (step velocity) than their non-faller counterparts (p<0.004). |
| | | Micro gait: Step Velocity (m/s), step length (m), swing time var (s), step velocity var (m/s), step length var (m), step time var (s), stance time var (s), step time (s), swing time (s), stance time (s), step time asy (s), swing time asy (s), stance time asy (s), and step length asy (m) | | | |
| **Galperin et al. (2019) [51]** | PD motor symptoms analysis | Gait quantity (i.e., number of steps and number of walking bouts) and gait quality (i.e., step length (m), step regularity, and the amplitude of dominant frequency ($g^2$/Hz)) | Walk detection algorithm | **Controlled environment** | Demographics and subject characteristics, laboratory-based measures of gait symmetry, and motor symptom severity together explained the 27.1% of the variance in total daily-living physical activity |
| | | | | **Home** | |
| **Heijmans et al. (2019) [52]** | Detection of tremor severity | Tremor episodes (starting time and duration) | Linear regression | **Home** | Tremor severity outcome (classifier) and tremor score diary had correlations of up to r = 0:43 |
| | | Tremor severity score | | | |
| **Mantri et al. (2019) [53]** | Monitoring of physical activity in PD patients and its correlation with Physical Activity Scale in the Elderly | Moderate-vigorous physical activity (min/day), number of steps | Algorithm for level of physical activity | **Home** | Median moderate-vigorous physical activity was 8.1 min/day and not correlated with Physical Activity Scale in the Elderly (ρ = -0.003, p = 0.98). |
| **McNames et al. (2019) [54]** | Detection of tremor episodes | Tremor episodes (starting time and duration) | Walk detection algorithm Tremor estimation: using thresholds (frequency analysis | **Home** | In the control cohort, the algorithm detected tremor incorrectly 1.1% of the time or less. Moreover, there was a good correspondence between constancy of rest tremor as measured and UPDRS (ρ = 0:54). |

Abbreviations: AIM = abnormal involuntary movements, ANOVA = Analysis of variance, ANN = Artificial Neural Network, asy = asymmetry, CV = Coefficient of Variation, DD = Diverse Density, EM-DD = Expectation Maximization version of Diverse Density, ICC = Intra Class Correlation, ID-APR = discriminative variant of the axis-parallel hyper-rectangle, kNN = k-Nearest Neighbor, MI-SVM = Multiple Instance Support Vector Machine, ML = medio-lateral PD = Parkinson's Disease, SVM = Support Vector Machine, UPDRS = Unified Parkinson's Disease Rating Scale, var = variability.

activity [53]. Ten articles examined symptoms, side-effects of treatments, and their fluctuations, including two on bradykinesia [32, 35], four on tremor [34, 47, 52, 54], four on dyskinesia [34, 35, 45, 46] and four on the on/off state [41, 45, 46, 49] (Table 2). During gait impairment monitoring, sensors were typically placed at the lower back, in 63.6% and 57.1% of the works taking place in the lab or at-home, respectively [33, 38, 39, 42–44, 48, 50, 51].

Lower back sensors were also combined with IMUs at the top of each foot (lab: 9.1%, home: 14.3%) [38, 40], waist (lab: 18.2%, home: 21.4%) [36, 37, 53], and left shank (lab: 9.1%, home: 7.1%) [31]. To monitor symptoms and their fluctuations, typical sensor positions included the waist (lab: 25%, home: 20%) [41, 49], wrist (lab: 25%, home: 20%) [35, 46, 47], both wrists (lab: 25%, home: 30%) [45, 52, 54] or in a combination of both ankles and the waist (lab: 25%, home: 20%) [32, 34] (Table 2).

## Aims, outcome measures, and types of analysis

Fourteen articles investigated gait impairments with the aim of assessing different mobility tendencies and habits in daily life (Tables 2 and 3). Kinematics [31, 33, 36–38, 40, 43, 44], also in combination with frequency measures [33, 39, 42, 51] were computed to study PD and healthy subjects [31, 33, 36–38, 40, 43, 44, 53], or different PD populations such as recently and previously diagnosed patients [31], fallers and non-fallers [39, 50], and subjects with or without freezing-of-gait [42, 48, 50]. In order to extract kinematic and frequency parameters, walk detection algorithms were implemented in seven cases [31, 36, 37, 39, 42, 44, 50, 51], while turning algorithms in three [38, 40, 48].

Ten articles studied symptoms and their fluctuations with the intention of detecting brady-kinesia [32, 35, 41, 49], tremor [34, 47, 52, 54], dyskinesia [34, 35, 41, 45, 46, 49], and on/off state episodes [41, 45, 46, 49]. Supervised machine learning approaches, such as Artificial Neural Networks (ANN) [45], Fuzzy logic algorithms [35], linear regression [52] and Support Vector Machine (SVM) [32] models were used in this context. One publication used multiple instance learning algorithms [34], namely, the Diverse Density (DD), Expectation Maximization version of Diverse Density (EM-DD), Discriminative variant of the axis-parallel hyper-rectangle (ID-APR), Multiple instance learning k-Nearest Neighbor (MIL-kNN) and Multiple Instance Support Vector Machine (MI-SVM). Finally, four studies used thresholds and analyses of frequency patterns [41, 47, 49, 54]. Walk [41, 49, 54] and activity recognition [32] algorithms were also employed in order to assess symptoms during specific patients' actions (Tables 2 and 3).

## Results of the included studies

Yoneyama et al. (2013/2014) [36, 37] found that the average duration of the gait cycle was longer in PwP ($1.16 \pm 0.20$ s) compared to controls ($1.08 \pm 0.19$ s; $p < 0.001$). Similarly, Del Din et al. (2016) [44] reported that Parkinsonians walked with slower and shorter steps (step velocity:$1.038 \pm 0.422$ m/s and $1.103 \pm 0.411$ m/s for PD and controls, respectively; $p < 0.001$). Moreover, PwP presented less consistent (e.g. step time variability: $0.175 \pm 0.156$ s for control and $0.181 \pm 0.179$ for PD; $p = 0.07$) and asymmetric (e.g. step time asymmetry: $0.093 \pm 0.086$ for control and $0.098 \pm 0.142$ for PD; $p = 0.116$) walking patterns [44], with fluctuations in kinematics and frequency measures compared to healthy subjects [31, 33, 44].

Three studies also investigated turning [38, 40, 48] and confirmed that PwP take shorter turns (2.0 s and 2.2 s for PD and control, respectively; $p = 0.001$) with smaller angles (92.0˚ and 95.2˚ for PD and control, respectively; $p = 0.001$) [38]. In addition, PwP completed the turning movement at a slower pace than controls (turn mean velocity: $38 \pm 5.7$˚/s and $43.3 \pm 4.8$˚/s, respectively; $p = 0.04$) and with a greater number of steps (mean number of steps: $3.2 \pm 0.8$ and $1.7 \pm 1.1$, respectively; $p = 0.04$) [40].

One publication investigated the correlation of the monitored overall steps taken (3615/day) and time spent in moderate-to-vigorous-physical-activities (MVPA, 8.1 min/day) with the self-reported activity using the Physical Activity Scale in the Elderly–PASE; there was a moderate correlation for steps ($r = 0.56$, $p = 0.003$), but practically no correlation for MVPA (r

= -0.003, p = 0.98) [53]. Finally, two works estimated that falls occurred most frequently in PwP with a more variable, less consistent walking pattern [39, 50]; furthermore frequency sensor-derived measures were successfully able to predict future falls even in patients with no previous fall history [39].

When assessing symptoms at-home, Pastorino et al. [32] classified bradykinesia with respect to the UPDRS outcome as measured by clinicians twice per day and achieved an accuracy of 68.3 ± 8.9% with the standard SVM and 74.4 ± 14.9% with a meta-analysis algorithm. Das et al. [34] obtained an accuracy versus symptom diaries of over 90% for both dyskinesia and tremor detection with a multiple instance learning ID-APR classifier. During a recording of ten days, a significant correlation (p < 0.0005) with an r = 0.64 between global median bradykinesia and UPDRS, and a correlation (p < 0.05) with a margin of error of 3.9 (over a range 0–8) between global median dyskinesia and UPDRS was found by Griffiths et al. [35]. Pérez-López et al. [41] developed an algorithm for the on/off state events recognition based on threshold detection and analysis of frequency patterns with a sensitivity of 99.9% and a specificity of 99.9% (compared to the symptom diary). Rodriguez-Molinero et al. [49] has built upon the previous study, increasing the sample size to 23 PwP and achieving an accuracy of 92.20%. Fisher et al. [45] built an ANN classifier that was validated from symptom diaries with a sensitivity ranging from 38% to 52% and specificity from 83% to 93% for the on/off states and for dyskinesia. The method implemented by Ossig et al. [46] had a moderate-to-strong correlation with subject diaries for on/off states and dyskinesia (p-values ranging from 0.404 to 0.658). For the tremor assessment, Battista and Romaniello et al. [47] accomplished a sensitivity of 99.3%, a specificity of 99.6%, and an accuracy of 98.9% as against the tremor diaries; Heijmans et al. [52] reported correlations of up to r = 0.43, when compared to diaries, while McNames et al. [54] detected tremor presence (incorrectly) just 1.1% of the time or less in healthy volunteers.

## Discussion

The main aim of the present work is to review and compare previous studies on the monitoring of PwP using only wearable inertial sensors and with at least one data capture carried out during unsupervised home activities. The intent was to inform future works in which the authors aim to use body-fixed-sensors for extended periods of time in scenarios where data captures are not monitored either directly or via a videotape.

As a matter of fact, the evaluation of PD requires extensive judgement from highly-trained professionals, yet clinical assessments in a clinical setting provide only a partial overview of the disease's pathological progression [55]. In addition, numerous episodes related with PD are challenging to detect during laboratory-based short-term observations. To consistently analyse motor symptoms, fluctuations and gait impairments, long observation windows are required due to the complexity and sporadicity of such events [21].

Wearable motion sensors are able to monitor PwP outside of standard clinical environments (for example, in private homes or community dwellings), and provide technically and clinically relevant information for clinicians and patients; therefore, a continuous assessment of the pathology may improve the quality of life of PwP, allowing them to preserve their independence and avoid additional disease complications. [12, 56, 57].

### Characteristics of the studies

For the purpose of gathering large datasets from IMUs recordings lasting from one to 14 days, the most frequently used off-the-shelf devices were the DynaPort, Opal and AX3 (Fig 2), while five works used inertial non-commercial prototypes. The majority of the studies adopted off-the-shelf devices and off-line algorithm solutions. However, a potential implementation of ad-

hoc hardware and on-board algorithms could enhance real-time feedbacks and ultimately have a meaningful impact in the life of patients living, for instance, in rural communities and remote areas [35, 46]. In both cases, the direct manipulation of raw data, gathered during the free-living acquisitions, avoids the use of aggregated data (i.e. step, distance) generated by "black box" software of commercial devices.

In the reviewed articles, diaries were completed by PwP or caregivers in order to track daily activities, medication intake, and symptom occurrences. However, the use of self-report for a complex task, such as the self-detection and recording of motor status over a prolonged period, may lead to misinterpretations and errors, particularly in PwP who have impaired cognition [58]. Patients may not always be able to correctly identify their own motor fluctuations and symptoms or they may log motor symptoms in incorrect time slots, or forget to update the records and then complete them many hours later from a recalled general state of function. Reportedly, diaries are not a reliable means of comparison; for example, Erb et al. [58] found that 38% of PwP in this study omitted approximately 25% of entries. However, developing digital versions, with alerts and prompts, may lessen the drawbacks typically associated with traditional paper-based diaries for PwP [59], while the involvement of caregivers trained in the data collection could benefit the quality of the reports.

The number of subjects involved in the data collections is another important aspect with an impact on the results. Sample sizes varied considerably among studies and ranged from one [33, 52] to 170 PwP, [50] from one [33] to 172 [50] controls, and from 1 [52] to 342 [50] volunteers in total (PwP and controls) in unsupervised environments. No pre-study calculation was reported in any of the papers to justify the sample size chosen. As a consequence, the small number of volunteers in certain experimental protocols generated less conclusive and decisive results in terms of statistical power.

Devices' number and placement were various, depending on the outcomes measured. Concerning impaired locomotion, the center of mass was extensively used in literature to measure movement performance and level of stability [60–62]. Accordingly, to monitor activities such as walking and turning, most of the papers agreed to adopt a single sensor worn close to the waist [36, 37, 53] and lower back [33, 38, 39, 42–44, 48, 50, 51]. Besides, PwP may exhibit asymmetric walk due to the different level of impairment of the lower limbs, characterized by a reduction in walking speed, shuffling steps, and limited foot lifting [3]. Consequently, a sensor attached on the single limb would capture recordings with large variations in gait patterns and it would give just a partial overview of the patient's status.

Sensor positioning and number is also crucial for the assessment of multiple symptoms on different subjects. In fact, tremor, dyskinesia, bradykinesia, and other PD related motor fluctuations affect upper and lower limbs differently depending on the manifestation and stage of the disease [3]. Thus, a combination of several devices might be more suitable for multiple and concurrent evaluations, however this would compromise the comfort of the system. Yet, given that fewer wearable devices enhance the acceptability, wearability and usability of the system, a sensor on the wrist may offer a good trade-off between applicability and end-user convenience.

Finally, given the potential continuous long-term adoption of wearable systems by PwP, aspects which were neglected in the identified papers, such as a system's comfort of use, set-up process, instructions for use, support, aesthetics and display, should always be considered to guarantee long-term acceptability and efficacy of the system. For instance, the FDA-approved Parkinson's Kinetigraph system (PKG), which provides continuous, objective, ambulatory assessments of PD symptoms, has been proved to show high patient acceptability, with 81% of the users reporting satisfactory outcomes [63]. These considerations are crucial if the final purpose is to gather large datasets and if PwP have to interact on a daily basis with the system.

### Aim, outcome measures, type of analyses, and results

Kinematic parameters, such as duration of gait cycle, step length, and velocity, were clearly differentiated between the PD and healthy populations. In fact, PwP walked slower and with shorter steps [36, 37, 44]. Less consistent gait patterns with major fluctuations in kinematics and frequency were also observed [31, 33, 44]. Findings also underlined differences in turning [38, 40, 48], showing patients taking shorter turns with smaller angles and completing the turning movement slower and with a greater number of steps. Concerning the risk of falling, the relationship between the level of activity and impairments is still a matter of debate among the scientific community. On one side, more active patients could be more susceptible to falls since they are exposed to more unsafe situations, but on the other hand they could be at a lower risk of falling due to a better general health condition. Two reviewed articles estimated that falls occurred significantly more frequently in PwP with a less consistent walking pattern [39, 50], while fallers seemed to have a reduced capability to regulate gait due to a partial loss of postural stability [64]. Inertial wearable device can detect such impaired walking patterns and predict future falls even in patients with no previous fall history [39].

To evaluate tremor at-home, two papers reported an accuracy against the symptom diary higher than the 90% [34, 47]. In particular, Battista and Romaniello et al. [47] presented a promising method based on the spectral analysis of inertial data from a single wrist worn sensor, in conjunction with the detection of specific movement patterns generally related with Parkinsonism. To assess bradykinesia and dyskinesia, Griffiths et al. [35] implemented a fuzzy logic approach using data collected from an accelerometer on the most affected wrist; these algorithms are the core of the PKG, the first FDA-approved device for the continuous assessment of PD symptoms. In addition, regarding dyskinesia, Fisher et al. [45] developed an ANN classifier that was validated from symptom diaries obtaining a promising level of specificity (93%) but still with a low sensitivity level (38%). Finally, to detect on/ off episodes, Pérez-López et al. [41] and Rodriguez-Molinero et al. [49] developed an algorithm based on the extraction of gait features from an accelerometer on the waist. The algorithm showed an accuracy of 92.2% when compared to the results of the diaries, however this approach relied upon gait parameters and required patient's movement; therefore, it might not be suitable for the recognition during the advanced stage of the disease when PwP are mostly inactive.

### Conclusion

The systematic review included 24 studies on the monitoring of PD using inertial sensors during unsupervised home activities. Previous articles already underlined how the well-know "Hawthorne observation effect" [20] could influence the reliability of data gathered in a laboratory setting since participants perform better when completing scripted tasks and while observed by a clinician. Furthermore, episodes associated with PD usually require long periods of observation because of their complexity (i.e. the on/off phenomenon) or rarity (i.e. freezing of gait phenomenon). As a consequence, home based data captures could generate more complete and exhaustive results in the analysis of the Parkinson's disease.

Fourteen articles focused on postural and gait disturbances [31, 33, 36–40, 42–44, 48, 50, 51, 53] with the intention of evaluating mobility in daily life. The majority of the studies agreed that a position close to the center of mass (waist or lower back) was ideal for impaired gait analysis. Kinematic parameters, such as duration of gait cycle, step length, and velocity, were shown to be capable of discriminating PD and healthy subjects. Furthermore, researchers reported less consistent gait patterns in patients that may be used to predict falls in the Parkinsonian population [39].

Ten articles investigated symptoms and their fluctuations aiming to detect bradykinesia, tremor, dyskinesia, and on/off state episodes [32, 34, 35, 41, 45–47, 49, 52, 54]. Even if researchers were able to achieve accuracies over 90% in a free-living environment [34, 41, 47, 49], the assessment of multiple symptoms on different subjects necessitated the employment of a high number of wearable devices, compromising the user-friendliness of the system and patients' comfort. The wrist position may offer the best compromise between performance, applicability, and end-user convenience.

In conclusion, future studies commencing an assessment of PwP for prolonged time periods may look into the a) development and testing of dedicated hardware and software for real-time feedback that would also permit the interaction between clinicians and patients, and b) the incorporation of digital versions of diaries with alerts and prompts in the study's design that would allow the correlation between quantitative measurements and self-reported outcomes. Additionally, characteristics which were ignored by researchers, such as the system's comfort of use, set-up process, instructions for use, support, aesthetics and display, need to be strongly considered. These reflections are fundamental for the efficacy of a health care system that will be used mostly by older people in a social environment and it should not affect patients physically or psychologically [12, 56, 57, 65–70].

## Supporting information

**S1 Table. Risk of bias.**
(DOCX)

**S2 Table. PRISMA checklist.**
(DOC)

**S3 Table. Review assessment tool.**
(XLSX)

## Author Contributions

**Conceptualization:** Marco Sica.

**Data curation:** Marco Sica.

**Formal analysis:** Marco Sica.

**Funding acquisition:** Salvatore Tedesco, Suzanne Timmons, John Barton, Brendan O'Flynn.

**Investigation:** Marco Sica, Salvatore Tedesco, Colum Crowe.

**Methodology:** Marco Sica.

**Supervision:** Dimitrios-Sokratis Komaris.

**Visualization:** Marco Sica.

**Writing – original draft:** Marco Sica.

**Writing – review & editing:** Salvatore Tedesco, Colum Crowe, Lorna Kenny, Kevin Moore, Suzanne Timmons, John Barton, Brendan O'Flynn, Dimitrios-Sokratis Komaris.

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
