## [Decision Letter · Decision Letter 0]

28 Sep 2020

PONE-D-20-24431

Continuous home monitoring of Parkinson’s disease using inertial sensors: a systematic review

PLOS ONE

Dear Dr. Sica,

Thank you for submitting your manuscript to PLOS ONE. After careful consideration, we feel that it has merit but does not fully meet PLOS ONE’s publication criteria as it currently stands. Therefore, we invite you to submit a revised version of the manuscript that addresses the points raised during the review process.

Dear Authors,

Both reviewers indicated major revision for your paper. You may address all comments and suggestions of the reviewers. The main concerns are about the paper justification of the statements, rationale of purpose and methodology description. The concerns are important for the reviewers to consider the paper for publication in the Journal.

We look forward to receiving your revised manuscript.

Kind regards,

Fabio A. Barbieri, PhD

Academic Editor

PLOS ONE

Journal Requirements:

2. Detailed reasons for exclusion are missing from the PRISMA flowchart. Please modify your PRISMA flowchart to address this issue.

'This project is co-funded by the European Regional Development Fund (ERDF) under Ireland’s European Structural and Investment Funds Programme 2014-2020. Aspects of this work have been supported in part by INTERREG NPA funded project SenDOC. Aspects of this publication were supported by Enterprise Ireland and Abbvie Inc. under grant agreement no. IP 2017 0625.'

We note that you received funding from a commercial source: Abbvie Inc

**Please include this amended Role of Funder statement in your cover letter; we will change the online submission form on your behalf.**

**Please include your amended Competing Interests Statement within your cover letter. We will change the online submission form on your behalf.**

Reviewers' comments:

Reviewer's Responses to Questions

**Comments to the Author**

1. Is the manuscript technically sound, and do the data support the conclusions?

Reviewer #1: Partly

Reviewer #2: Yes

2. Has the statistical analysis been performed appropriately and rigorously? 

Reviewer #1: Yes

Reviewer #2: N/A

3. Have the authors made all data underlying the findings in their manuscript fully available?

Reviewer #1: Yes

Reviewer #2: Yes

4. Is the manuscript presented in an intelligible fashion and written in standard English?

Reviewer #1: No

Reviewer #2: Yes

5. Review Comments to the Author

Reviewer #1: ABSTRACT

“Parkinson’s disease (PD) is a progressive neurological disorder of the central nervous system 17 that primarily affects motor functions” This statement could be modified, since it is acknowledged that PD is not only affecting the motor function, but it also presents a large diversity of non-motor symptoms.

“In fact, 23 home free-living data collection is more relevant to assessing PD status as it affects the person, 24 capturing motor function during daily living” This statement is wrongly justified and incorrect.

“In summary, inertial sensors are capable of gathering data over a long period of 29 time and provide relevant information” I would argue these are the main conclusions of the literature review performed.

INTRODUCTION

“affecting approximately 3% of the population over the 39 age of 65 and 10% over the age of 80 [1].” Consider an update of the reference and a correction of the epidemiology stated

“The symptoms of PD are multiple, with many being 40 related to motor dysfunction, such as tremor, rigidity, bradykinesia, and to postural instability 41 [2].” This statement is vague, consider indicating the relationship of the mentioned symptoms to the diagnose of PD and the traditional consideration of PD as a motor disease

“Tremor is common, typically occurring unilaterally and later becoming bilateral with

43 disease progression [2].”

Resting tremor: rhythmic oscillatory involuntary movement occurring in approximately 75% of patients with PD. It appears when the affected body part is relaxed and the action of gravitational forces is not counteracted, particularly in distal parts of the extremities. Resting tremor is presented with a frequency between 3 and 6 Hz and a variable amplitude, from less than 1 cm to > 10 cm. At the onset of the disease, patients typically present asymmetrical tremors, which are more prominent in the upper extremities, equally affecting both sides of the body as the disease progresses.

“The “on” and “off” 52 phenomenon, in Levodopa-treated patients, describes motor fluctuations that occur as the levels 53 of dopamine in the brain drop, followed by a worsening of the motor function: during the "on" 54 state the symptoms are well managed, while in the "off" state they deteriorate.” Consider indicate the duration of the on phase and the daily fluctuation effect.

Line 57 – 69. Consider indicating the fact that UPDRS is also time demanding, not routinely performed, unreliable and qualitative assessment.

“Small-size, lightweight, low-power wearable motion sensors may aid in the objective

71 assessment of PD motor symptoms.” Why and how?

Line 78 – 90. Please justify this paragraph better with information from the following reference:

Long-term unsupervised mobility assessment in movement disorders.

Warmerdam E, Hausdorff JM, Atrsaei A, Zhou Y, Mirelman A, Aminian K, Espay AJ, Hansen C, Evers LJW, Keller A, Lamoth C, Pilotto A, Rochester L, Schmidt G, Bloem BR, Maetzler W.

The connection between the need of daily-life monitoring with IMUS and its clinical relevance should be better justified.

METHODS

Synonyms that could have been used for IMU: body-fixed-sensor, inertial measurement unit, sensor, monitor

RESULTS

Consider including a schema (PRISMA guidelines) on the specific exclusion flowchart.

Consider including a graph with the number of articles found for each specific year of the range

“Data was collected by accelerometers only (in 46.7% and 50% of the studies in the lab

165 or at-home, respectively)” This statement is confusing, were 46.7% of studies using an accelerometer only in the lab, and the 50% only at home?

“Fourteen works investigated activity 177 monitoring, eight of which focused on walking 178 [24, 26, 29, 30, 35-37, 44], three on turning [31, 33, 41], two on falls [32, 43] and one on 179 physical activity [46].” Were these articles not focused on monitoring gait impairments?

“and four on the on/off 181 state” since on/off status is not a PD symptom, it should be carefully stated in this sentence to avoid confusion

“Similarly, 218 Del Din et al. (2016) [37] reported PwP walked with slower and shorter steps” These findings are in opposition to the precedent statement

“Moreover, less consistent walking patterns with major fluctuation in kinematics and frequency 221 measures compared to healthy subjects were observed” This information is technically and clinically relevant and should be further developed. Consider referring to the PwP population as the subject of the sentence.

Table 3. Consider including certain technical information regarding the IMU sensors: sampling frequency, range, duration of assessment, etc.

DISCUSSION

“yet 283 clinical assessments in a clinic setting provide only a partial overview of the disease’s 284 pathological progression [48].” this is not the main aim of the literature review, nor the most common used of IMUs for PwP monitoring

“and hence the quality of life of PwP, allowing them to preserve their independence, avoid 288 additional disease complications and reduce personal” This statement is not carefully stated, nor justified

“it was also evident that the adoption of 300 ad-hoc hardware has led to the optimization of the on-board algorithms, enhancing the real301 time feedback given to the PwP and/or clinicians” This cannot be concluded based on the findings of this article

What is the benefit of long term monitoring? Is there any reliability in a longer measurement?

It could be mentioned the “black box” reality on using the software of commercial devices

Consider mentioning confounders and limitations on real-life monitoring of movements

Limitations on the use of questionnaires for real-life activity monitoring should is neither a main conclusion from this literature review. Did this literature review assess any correlation between quantitative measurements and self-reported outcomes?

“Sample sizes varied considerably among studies and ranged from one 317 [26, 45] to 170 [43] PwP and from one [26] to 172 [43] controls” What was the range of the larger sample size including PwP? What is the impact of a reduced sample size? Were any statistical analyses performed in these papers to justify the sample size chosen? Were there any clinical limitation?

“Concerning the devices’ position during activity monitoring (such as walking and

321 turning), most of the papers agreed that a site close to the center of mass such as the waist [29, 322 30, 46] and lower back [26, 31, 32, 35-37, 41, 43, 44] was ideal for impaired gait analysis.” Why? This needs to be biomechanically justified, since it is relevant for clinical implementation of the devices.

“consequently a sensor attached on the limbs would capture 325 recordings with large variations in gait patterns during activity monitoring.” This a wrong statement in the current state.

“manifest differently and on various locations from person to person [2].” due to laterality?

“should be always considered 336 to guarantee the long-term acceptability of the system.” for what purpose? This needs to be developed and justified

In general, this article needs an exhaustive revision, including a better justification of the statements. Moreover, the aim of this review is not well driven.

Reviewer #2: Wearable inertial sensors have the potential for passive monitoring of the Parkinson’s disease in the free-living environment. This review summarized the findings nicely and it is well written. For further refinement, I have following comments:

- Why the papers only from 2010 to 2020 were selected for review?

- Figure quality is very poor, difficult to read.

- Table 2 is not adding much to the manuscript as you already given detail in the section 3.2. You can put it in the supplementary material by highlighting the missing parts instead of just giving them score.

- Please discuss the specifications of the inertial sensors used in these studies. Also shed some light either these sensors are fit for this purpose, or there is a need of custom-built sensors for passive monitoring of the disease.

- Please revise line 284-288 for better readability.

- Why we need assessment of 7 days? Please discuss its importance.

- Please discuss in detail, what kind of features can give better discrimination between PwP and controls in the free-living environment and why? How these features are going to be different from the controlled environment?

- Please indicate the future directions in bullet form based on the summary of these 24 papers.

6. PLOS authors have the option to publish the peer review history of their article (what does this mean?). If published, this will include your full peer review and any attached files.

Reviewer #1: No

Reviewer #2: No

---

## [Author Response · Author response to Decision Letter 0]

25 Nov 2020

The authors would like to thank the editor and reviewers for their thorough review and encouraging comments, as we believe that their input has greatly improved our manuscript. We hope that the amendments listed here have positively addressed all their comments. Please also note that all changes in the manuscript can be tracked with word’s “track changes” feature. 

Editor 

Thank you for this comment. We have modified the manuscript and file naming in order to meet PLOS ONE's style requirements.

2) Detailed reasons for exclusion are missing from the PRISMA flowchart. Please modify your PRISMA flowchart to address this issue.

We have modified the PRISMA flow diagram in order to add the information requested. For your convenience, we have copied the PRISMA diagram bellow:

3) Thank you for stating the following in the Financial Disclosure section:

'This project is co-funded by the European Regional Development Fund (ERDF) under Ireland’s European Structural and Investment Funds Programme 2014-2020. Aspects of this work have been supported in part by INTERREG NPA funded project SenDOC. Aspects of this publication were supported by Enterprise Ireland and Abbvie Inc. under grant agreement no. IP 2017 0625.'

We note that you received funding from a commercial source: Abbvie Inc

**Please include this amended Role of Funder statement in your cover letter; we will change the online submission form on your behalf.** 

**Please include your amended Competing Interests Statement within your cover letter. We will change the online submission form on your behalf.**

Thank you for this comment. We have modified the section Funding as follows (lines 443-449):

“This project is co-funded by the European Regional Development Fund (ERDF) under Ireland’s European Structural and Investment Funds Programme 2014-2020. Aspects of this work have been supported in part by INTERREG NPA funded project SenDOC. Aspects of this publication were supported by Enterprise Ireland and Abbvie Inc. under grant agreement no. IP 2017 0625. The funders had no role in study design, data collection and analysis, decision to publish, or preparation of the manuscript”.

In addition, we have modified our cover letter accordingly:

“This manuscript has emanated from research supported by the European Regional Development Fund (ERDF) under Ireland’s European Structural and Investment Funds Programme 2014-2020. Aspects of this work have been supported in part by INTERREG NPA funded project SenDOC. Aspects of this publication were supported by Enterprise Ireland and Abbvie Inc. under grant agreement no. IP 2017 0625. The funders had no role in study design, data collection and analysis, decision to publish, or preparation of the manuscript”. 

We have also added a clearer Competing Interests Statement (at the end of the cover letter):

Competing Interests Statement: the funders had no role in study design, data collection and analysis, decision to publish, or preparation of the manuscript. There were no restrictions on sharing of data and/or materials.

Reviewer 1 

Abstract

1) “Parkinson’s disease (PD) is a progressive neurological disorder of the central nervous system that primarily affects motor functions” This statement could be modified, since it is acknowledged that PD is not only affecting the motor function, but it also presents a large diversity of non-motor symptoms.

Thank you for this comment. We have modified the sentence as follows (lines 14-17):

“Parkinson’s disease (PD) is a progressive neurological disorder of the central nervous system that deteriorates motor functions, while it is also accompanied by a large diversity of non-motor symptoms such as cognitive impairment and mood changes, hallucinations, and sleep disturbance”. 

2) “In fact, home free-living data collection is more relevant to assessing PD status as it affects the person, capturing motor function during daily living” This statement is wrongly justified and incorrect.

This sentence was misleading and we have deleted it, modifying the paragraph accordingly (lines 17-22):

“Parkinsonism is evaluated during clinical examinations and appropriate medical treatments are directed towards alleviating symptoms. Tri-axial accelerometers, gyroscopes, and magnetometers could be adopted to support clinicians in the decision-making process by objectively quantifying the patient’s condition. In this context, at-home data collections aim to capture motor function during daily living and unobstructedly assess the patients’ status and the disease’s symptoms for prolonged time periods”. 

3) “In summary, inertial sensors are capable of gathering data over a long period of time and provide relevant information” I would argue these are the main conclusions of the literature review performed. 

We have modified the sentence as follows (lines 28-34):

“In summary, inertial sensors are capable of gathering data over a long period of time and have the potential to facilitate the monitoring of people with Parkinson’s, providing relevant information about their motor status. Concerning gait impairments, kinematic parameters (such as duration of gait cycle, step length, and velocity) were typically used to discern PD from healthy subjects, whereas for symptoms’ assessment, researchers were capable of achieving accuracies of over 90% in a free-living environment”.

Introduction

4) “affecting approximately 3% of the population over the age of 65 and 10% over the age of 80 [1].” Consider an update of the reference and a correction of the epidemiology stated

We have updated the reference as follows (lines 44-46):

“Its incidence rises dramatically with age, affecting approximately 6.2 million people worldwide in 2015 [1]”.

Added reference:

Vos T, Allen C, Arora M, Barber RM, Bhutta ZA, Brown A, Carter A, Casey DC, Charlson FJ, Chen AZ, Coggeshall M. Global, regional, and national incidence, prevalence, and years lived with disability for 310 diseases and injuries, 1990–2015: a systematic analysis for the Global Burden of Disease Study 2015. The lancet. 2016 Oct 8;388(10053):1545-602.

5) “The symptoms of PD are multiple, with many being related to motor dysfunction, such as tremor, rigidity, bradykinesia, and to postural instability [2].” This statement is vague, consider indicating the relationship of the mentioned symptoms to the diagnose of PD and the traditional consideration of PD as a motor disease.

We have modified the introduction sentence as follows (lines 46-50).

 “The symptoms of PD are multiple, with the most identifiable being related to motor degeneration. In general, they appear gradually and become more evident with the worsening of the disease, varying from person to person. The diagnosis of PD can be challenging, especially at an early stage, due to the lack of specific tests [2]. The most recognizable symptoms include tremor, rigidity, bradykinesia, and postural instability [3]”. 

Added reference:

Tolosa E, Wenning G, Poewe W. The diagnosis of Parkinson's disease. The Lancet Neurology. 2006 Jan 1;5(1):75-86.

6) “Tremor is common, typically occurring unilaterally and later becoming bilateral with

disease progression [2]”

Resting tremor: rhythmic oscillatory involuntary movement occurring in approximately 75% of patients with PD. It appears when the affected body part is relaxed and the action of gravitational forces is not counteracted, particularly in distal parts of the extremities. Resting tremor is presented with a frequency between 3 and 6 Hz and a variable amplitude, from less than 1 cm to > 10 cm. At the onset of the disease, patients typically present asymmetrical tremors, which are more prominent in the upper extremities, equally affecting both sides of the body as the disease progresses.

Thank you for this comment. Rest tremor is the most typical type of tremor in early PD stages, affecting the 75% of PwP. However, postural and kinetic tremor can affect roughly half of these patients and can occur in absence of resting tremor. We have modified the introduction sentence as follows, providing a general definition of tremor (lines 51-53).

“Tremor typically appears at the distal part of the limbs, affecting a single arm or leg; it is more pronounced in the upper extremities and it progresses bilaterally with the degeneration of the disease”. 

7) “The “on” and “off” phenomenon, in Levodopa-treated patients, describes motor fluctuations that occur as the levels of dopamine in the brain drop, followed by a worsening of the motor function: during the "on" state the symptoms are well managed, while in the "off" state they deteriorate.” Consider indicate the duration of the on phase and the daily fluctuation effect.

Thank you for this comment. We have expanded the “on” and “off” phenomenon description, adding the duration of the on phase and the daily fluctuation effect (lines 61-68).

“The “on” and “off” phenomenon in Levodopa-treated patients, describes motor fluctuations that occur as the levels of dopamine in the brain drop, followed by a worsening of the motor function: during the "on" state the symptoms are well managed, while in the "off" state they deteriorate. In newly diagnosed people with Parkinson’s (PwP), the response to a single drug intake may last for several hours, whereas with the progression of the disease the drug’s effect is shortened (4 hours or less), and patients need to decrease intervals between doses and/or increase the dosages [5,6]”.

Added references:

Freitas ME, Hess CW, Fox SH. Motor complications of dopaminergic medications in Parkinson’s disease. InSeminars in neurology 2017 Apr (Vol. 37, No. 2, p. 147). NIH Public Access.

Anderson E, Nutt J. The long-duration response to levodopa: phenomenology, potential mechanisms and clinical implications. Parkinsonism & related disorders. 2011 Sep 1;17(8):587-92.

8) Line 57 – 69. Consider indicating the fact that UPDRS is also time demanding, not routinely performed, unreliable and qualitative assessment.

Thank you for this comment. We have highlighted that the UPDRS is time demanding (lines 76-77). In addition, we have highlighted that is a qualitative assessment and not routinely performed (lines 72). Finally, lines 77-80 explain why the UPDRS is partially unreliable.

“To ensure the appropriate medical treatment and correct dose of medication for an individual, PwP are infrequently evaluated with qualitative clinical assessments that are based on the subjective judgment of specialists, such as the Movement Disorder Society - Unified Parkinson’s Disease Rating Scale (MDS-UPDRS), or specifically for dyskinesia, the modified Abnormal Involuntary Movement Scale (m-AIMS) [8,9]. Yet, due to the heterogeneity and complexity of PD symptoms, such clinical assessments can be challenging and time consuming. Clinicians with different backgrounds and experiences might also vary in their interpretations of the MDS-UPDRS and m-AIMS [10]. Equally, a person’s motor state at a clinic appointment may not be typical of their usual state, enhanced by fatigue, dehydration from travelling or anxiety [10]”.

9) “Small-size, lightweight, low-power wearable motion sensors may aid in the objective assessment of PD motor symptoms.” Why and how?

Please see answer in comment 11.

10) Line 78 – 90. Please justify this paragraph better with information from the following reference:

Long-term unsupervised mobility assessment in movement disorders.

Warmerdam E, Hausdorff JM, Atrsaei A, Zhou Y, Mirelman A, Aminian K, Espay AJ, Hansen C, Evers LJW, Keller A, Lamoth C, Pilotto A, Rochester L, Schmidt G, Bloem BR, Maetzler W. Long-term unsupervised mobility assessment in movement disorders.

Reference added as suggested, also please see answer in comment 11 for more details.

11) The connection between the need of daily-life monitoring with IMUS and its clinical relevance should be better justified.

Thank you for this comment. We have modified the paragraph in the introduction to emphasize on the relevance of IMU long term monitoring and address the comments 9, 10, and 11 (lines 84-100). 

“Due to their small-size, light weight, and low-power, wearable motion sensors have already demonstrated their clinical relevance in healthcare [11–13] and daily-life monitoring [14,15]. The most widely used sensors are tri-axial accelerometers, gyroscopes, and magnetometers, commonly combined in an inertial measurement unit (IMU) that can capture three-dimensional orientation, and linear and angular velocities [16,17]. Thanks to the development of miniaturized hardware technologies capable of collecting and storing large amount of raw data [18], IMUs may offer the opportunity to improve the evaluation of the PD motor symptoms by collecting free-living movements for prolonged period of time outside the laboratory environment. Former studies, such as the one by Bolem et al. (2001) [19], have reported that PwP walk better when observed rather than when unsupervised in their daily lives. This is a consequence of the well-known “Hawthorne observation effect” [20]: free-living activities involve a combination of tasks with varying complexities, challenges and distractions that may reduce attention. In addition, numerous episodes related with PD are challenging to detect during laboratory-based observation because of their complexity (i.e. the on/off phenomenon) or rarity (i.e. freezing of gait phenomenon) [21]. As a consequence, a thorough evaluation of a PwP requires the data to be gathered during long observation windows while patients go ahead with normal every day activities”. 

Added references:

Warmerdam E, Hausdorff JM, Atrsaei A, Zhou Y, Mirelman A, Aminian K, Espay AJ, Hansen C, Evers LJ, Keller A, Lamoth C. Long-term unsupervised mobility assessment in movement disorders. The Lancet Neurology. 2020 May 1;19(5):462-70.

Amft O, Junker H, Troster G. Detection of eating and drinking arm gestures using inertial body-worn sensors. InNinth IEEE International Symposium on Wearable Computers (ISWC'05) 2005 Oct 18 (pp. 160-163). IEEE.

Mancini M, Schlueter H, El-Gohary M, Mattek N, Duncan C, Kaye J, Horak FB. Continuous monitoring of turning mobility and its association to falls and cognitive function: a pilot study. Journals of Gerontology Series A: Biomedical Sciences and Medical Sciences. 2016 Aug 1;71(8):1102-8.

Methods

12) Synonyms that could have been used for IMU: body-fixed-sensor, inertial measurement unit, sensor, monitor

We have made appropriate changes in lines 101, 186, 291, and 308:

“Previous reviews have already investigated monitoring of PD using body-fixed-sensors”.

“In five studies, volunteers wore prototype sensors”.

“The intent was to inform future works in which the authors aim to use body-fixed-sensors for ….”.

“… while five works used non-commercial prototypes”.

Results

13) Consider including a schema (PRISMA guidelines) on the specific exclusion flowchart.

Please see the answer in comment 14.

14) Consider including a graph with the number of articles found for each specific year of the range

We have modified the PRISMA flow diagram in order to add the information requested on both comments 13 and 14. The identified and included articles per year are mentioned in the first and last elements of the flow-diagram, respectively. We have also added exclusion criteria and the corresponding number of papers for each criterion. For your convenience, we copied the Prisma diagram bellow: 

15) “Data was collected by accelerometers only (in 46.7% and 50% of the studies in the lab or at-home, respectively)” This statement is confusing, were 46.7% of studies using an accelerometer only in the lab, and the 50% only at home?

Accelerometer data (without gyroscope or magnetometer) were collected in 46.7% of all lab studies and 50% of the studies collecting data at home. We have changed the sentence as follows (lines 177-180):

“Data were collected by accelerometers alone (in 46.7% of the studies recording in labs and 50% of the studies at-home) [33,35–37,41,44–47,49–51,53], in combination with gyroscopes (lab: 33.3%, home: 33.3%) [31–33,39,42,43,48,52] or along with gyroscopes and magnetometers (lab: 20 %, home: 16.7%) [34,38,40,51,54]”.

16) “Fourteen works investigated activity monitoring, eight of which focused on walking [24, 26, 29, 30, 35-37, 44], three on turning [31, 33, 41], two on falls [32, 43] and one on 179 physical activity [46].” Were these articles not focused on monitoring gait impairments?

The main goal of these studies was to report on different mobility tendencies and habits in daily life, and gait impairments during walking and turning and how those can lead to falls. We have substitute “activity monitoring” with “gait impairments” as follows (lines 25, 31, 189, 194, 207): 

“Fourteen investigated gait impairments, eight of which focused on walking, three on turning, two on falls, and one on physical activity”.

“Concerning gait impairments, kinematic parameters (such as duration of gait cycle, step length, and velocity) were typically …”.

“Fourteen works investigated gait impairments, eight of which focused on walking [31,33,36,37,42–44,51], three on turning [38,40,48], two on falls [39,50] and one on physical activity [53]”.

“During gait impairments monitoring, sensors were typically placed at the lower back, in 63.6% and 57.1% of the works taking place in the lab or at-home, respectively [33,38,39,42–44,48,50,51]”.

“Fourteen articles investigated gait impairments with the aim of assessing different mobility tendencies and habits in daily life (Table 2 and 3)”.

In addition, we have substitute “activity monitoring” with “gait impairments” in table 2

Author/Study Device Sampling frequency No of Sensors Sensors Sensors Position Assessment Subjects Diary Symptoms and Fluctuations Gait Impairments

 Accelerometer Gyroscope Magnetometer Patients Controls Activities Medication Symptoms Bradykinesia Tremor On/Off State Dyskinesia Walking Turning Fall Detection Physical Activity

17) “and four on the on/off state” since on/off status is not a PD symptom, it should be carefully stated in this sentence to avoid confusion

Thanks to underlining this, we have modified the text in order to avoid misunderstandings (lines 191-192, 198, 215, 409). 

“Ten articles examined symptoms, side-effects of treatments, and their fluctuations, including two on bradykinesia [32,35], four on tremor [34,47,52,54], four on dyskinesia [34,35,45,46] and four on the on/off state [41,45,46,49] (Table 2)”.

“To monitor symptoms and their fluctuations, typical sensor positions included …”.

“Ten articles studied symptoms and their fluctuations with the intention of detecting …”.

“Ten articles investigated symptoms and their fluctuations aiming to detect bradykinesia, tremor, dyskinesia, and on/off state episodes [32,34,35,41,45–47,49,52,54]”.

In addition, we have substitute “symptoms monitored” with “symptoms and fluctuation” in table 2

Author/Study Device Sampling frequency No of Sensors Sensors Sensors Position Assessment Subjects Diary Symptoms and Fluctuations Gait Impairments

 Accelerometer Gyroscope Magnetometer Patients Controls Activities Medication Symptoms Bradykinesia Tremor On/Off State Dyskinesia Walking Turning Fall Detection Physical Activity

18) “Similarly, Del Din et al. (2016) [37] reported PwP walked with slower and shorter steps” These findings are in opposition to the precedent statement.

Thank you for this comment. The paragraph (in the original manuscript) “As expected, Yoneyama et al. (2013/2014) [29, 30] found that the average gait cycle was longer in PwP (1.16 ± 0.20 s) compared to controls (1.08 ± 0.19 s) (p < 0.001). Similarly, Del Din et al. (2016) [37] reported PwP walked with slower and shorter steps (step velocity 1.038 ± 0.422 m/s and 1.103 ± 0.411 m/s for PD and controls, respectively. p < 0.001)” underlines the differences in impaired walking. In particular, Yoneyama et al. (2013/2014) reported that the average gait cycle was longer in duration (s) in PwP compared to controls. The findings of Del Din et al. (2016) are also in agreement since the PwP walked slower and thus had longer gait cycles. 

To better clarify this, we have changed the paragraph as follows, (lines 229-233):

“Yoneyama et al. (2013/2014) [36,37] found that the average duration of the gait cycle was longer in PwP (1.16 ± 0.20 s) compared to controls (1.08 ± 0.19 s; p < 0.001). Similarly, Del Din et al. (2016) [44] reported that Parkinsonians walked with slower and shorter steps (step velocity:1.038 ± 0.422 m/s and 1.103 ± 0.411 m/s for PD and controls, respectively; p < 0.001)”.

19) “Moreover, less consistent walking patterns with major fluctuation in kinematics and frequency measures compared to healthy subjects were observed” This information is technically and clinically relevant and should be further developed. Consider referring to the PwP population as the subject of the sentence.

Thank you for emphasizing on this. We have extended the statement adding further insight on Step Time Variability and Step Time Asymmetry (lines 233-236).

“Moreover, PwP presented less consistent (e.g. step time variability: 0.175 ± 0.156 s for control and 0.181 ± 0.179 for PD; p = 0.07) and asymmetric (e.g. step time asymmetry: 0.093 ± 0.086 for control and 0.098 ± 0.142 for PD; p = 0.116) walking patterns [44], with fluctuations in kinematics and frequency measures compared to healthy subjects [31,33,44]”. 

20) Table 3. Consider including certain technical information regarding the IMU sensors: sampling frequency, range, duration of assessment, etc.

• We have included information on the devices’ sampling frequency (4th column) in table 3. A portion of the updated table is shown below for convenience.

Author/Study Device Sampling frequency No of Sensors Sensors Sensors Position Assessment Subjects Diary Symptoms and Fluctuations Gait Impairments

 Accelerometer Gyroscope Magnetometer Patients Controls Activities Medication Symptoms Bradykinesia Tremor On/Off State Dyskinesia Walking Turning Fall Detection Physical Activity

Yoneyama et al. (2013/2014) 

(Part 1 and 2) [29,30] Controlled environment Mimamori-gait system 100 Hz 1 ✓ - - Waist

(front centre) 3 lab sessions - 11 - - - ✓ 

 - 1 

 12 - 

 Home Mimamori-gait system 100 Hz 1 ✓ - - Waist

(front centre) 1 day 17 10 ✓ - - 

El-Gohary et al. (2014) [31]

Controlled environment Opal 128 Hz 1 ✓ ✓ ✓ Lower back 1 lab session 21 19 - - - ✓ ✓ 

 Home Opal 128 Hz 3 ✓ ✓ ✓ Lower back

On top of each foot 7 days 12 18 - - - 

• Technical information such as range, size, weight, etc:

We kindly believe that such technical information would significantly increase the size of the tables and compromise their readability. However, we have provided (when possible) the names of the devices used (column 2, table above) and therefore the readers could easily retrieve such technical information online. 

• Duration of assessment:

Frequently, authors only report on the number of days or lab sessions during which they followed patients (table above), while the exact duration of each assessment is not always reported. We do believe that is an important part of the methodology, especially for the reproducibility of the protocol. For this reason, we decided to mention this in the Results (lines 165-171): 

“Among the 24 identified papers, 15 included sessions that were recorded both at home and in the lab during one [33,35,38–45,51] , two [31,32], or three visits [36,37]; these studies included an initial calibration/validation in a supervised environment for the development of ad-hoc algorithms and then subsequent implementation/testing in an unsupervised setting. Besides the number of testing days and lab sessions, the exact duration of each lab assessment was not always specified, thus hindering the reproducibility of the protocol. In contrast, nine articles described data collection that was exclusively undertaken at home [34,46–50,52–54]”.

Discussion

21) “yet clinical assessments in a clinic setting provide only a partial overview of the disease’s pathological progression [48].” this is not the main aim of the literature review, nor the most common used of IMUs for PwP monitoring

We have made changes to better state the main aims of the review, please see the answer in the comment 22.

22) “and hence the quality of life of PwP, allowing them to preserve their independence, avoid additional disease complications and reduce personal” This statement is not carefully stated, nor justified

Thank you for this comment. Changes were made in the first paragraphs of the discussion should to address both comments 21 and 22. We have decided to emphasize on the importance of the continuous assessment of PD. In particular (lines 288-303):

“The main aim of the present work is to review and compare previous studies on the monitoring of PwP using only wearable inertial sensors and with at least one data capture carried out during unsupervised home activities. The intent was to inform future works in which the authors aim to use body-fixed-sensors for extended periods of time in scenarios where data captures are not monitored either directly or via a videotape. 

As a matter of fact, the evaluation of PD requires extensive judgement from highly-trained professionals, yet clinical assessments in a clinical setting provide only a partial overview of the disease’s pathological progression [55]. In addition, numerous episodes related with PD are challenging to detect during laboratory-based short-term observations. To consistently analyse motor symptoms, fluctuations and gait impairments, long observation windows are required due to the complexity and sporadicity of such events [21].

Wearable motion sensors are able to monitor PwP outside of standard clinical environments (for example, in private homes or community dwellings), and provide technically and clinically relevant information for clinicians and patients; therefore, a continuous assessment of the pathology may improve the quality of life of PwP, allowing them to preserve their independence and avoid additional disease complications. [12,56,57]”.

23) “it was also evident that the adoption of ad-hoc hardware has led to the optimization of the on-board algorithms, enhancing the real time feedback given to the PwP and/or clinicians” This cannot be concluded based on the findings of this article

We would like to thank the reviewer for the present comment. The majority of the studies used off-line solutions and off-the-self devices. This could be seen as limitation considering the final aim of the adoption of body-fixed-sensors to monitor PwP. We believe that the implementation of ad-hoc hardware and on-board algorithms could enhance real-time feedbacks given to the PwP and/or clinicians. For this reason, we have modified the paragraph as follows (lines 308-312):

“The majority of the studies adopted off-the-shelf devices and off-line algorithm solutions. However, a potential implementation of ad-hoc hardware and on-board algorithms could enhance real-time feedbacks and ultimately have a meaningful impact in the life of patients living, for instance, in rural communities and remote areas [35,46]”.

24) It could be mentioned the “black box” reality on using the software of commercial devices

The off-the-shelf devices adopted were loggers able to collect inertial raw data for extended periods of time. To our best knowledge the Parkinson’s Kinetigraph system is the only FDA approved device for the assessment of PD symptoms (lines 354-357).

“For instance, the FDA-approved Parkinson’s Kinetigraph system (PKG), which provides continuous, objective, ambulatory assessments of PD symptoms, has been proved to show high patient acceptability, with 81% of the users reporting satisfactory outcomes [63]”. 

And also (lines 380-383)

“To assess bradykinesia and dyskinesia, Griffiths et al. (2012) [35] implemented a fuzzy logic approach using data collected from an accelerometer on the most affected wrist; these algorithms are the core of the PKG, the first FDA-approved device for the continuous assessment of PD symptoms.”.

However, accordingly with the comment received; we have changed the paragraph below as follow (lines 308-314)

“The majority of the studies adopted off-the-shelf devices and off-line algorithm solutions. However, a potential implementation of ad-hoc hardware and on-board algorithms could enhance real-time feedbacks and ultimately have a meaningful impact in the life of patients living, for instance, in rural communities and remote areas [35,46]. In both cases, the direct manipulation of raw data, gathered during the free-living acquisitions, avoids the use of aggregated data (i.e. step, distance) generated by “black box” software of commercial devices”. 

25) What is the benefit of long term monitoring? Is there any reliability in a longer measurement?

Long-term monitoring is fundamental for reliable analysis of the Parkinson’s pathology. In fact, motor symptoms are too complex to be analysed short-term in a laboratory-based data capture. We have added the following paragraph (lines 295-298) 

“In addition, numerous episodes related with PD are challenging to detect during laboratory-based short-term observations. To consistently analyse motor symptoms, fluctuations and gait impairments, long observation windows are required due to the complexity and sporadicity of such events [21]”.

26) Consider mentioning confounders and limitations on real-life monitoring of movements

Please see the answer in the comment 32.

27) Limitations on the use of questionnaires for real-life activity monitoring should is neither a main conclusion from this literature review. Did this literature review assess any correlation between quantitative measurements and self-reported outcomes?

We would like to thank the reviewer for this comment. The diaries play an important role in PD research that aims to estimate motor symptoms. In fact, patients’ self-assessment was used before for the implementation of supervised machine learning algorithms as well as for the evaluation of their accuracy. Even though the present review didn’t assess any correlation between quantitative measurements and self-reported outcomes, the use of diaries for a complex task, such as the self-detection and recording of motor status over a prolonged period (in PwP population), is an intrinsic limitation of the reviewed studies. Considering the main intent of the present review, we kindly believe that we couldn’t ignore this aspect in order to give clearer directions to the researchers that aim to evaluate PD motor symptoms using IMUs for extended periods of time in an unsupervised environment.

28) “Sample sizes varied considerably among studies and ranged from one [26, 45] to 170 [43] PwP and from one [26] to 172 [43] controls” What was the range of the larger sample size including PwP? What is the impact of a reduced sample size? Were any statistical analyses performed in these papers to justify the sample size chosen? Were there any clinical limitations?

The works with a small number of volunteers generate less conclusive and decisive results in terms of statistical power. In addition, no analysis was conducted in any of the studies to justify the sample size chosen. We have decided to modify the paragraph as below: (lines 327-333).

“The number of subjects involved in the data collections is another important aspect with an impact on the results. Sample sizes varied considerably among studies and ranged from one [33,52] to 170 PwP, [50] from one [33] to 172 [50] controls, and from 1 [52] to 342 [50] volunteers in total (PwP and controls) in unsupervised environments. No pre-study calculation was reported in any of the papers to justify the sample size chosen. As a consequence, the small number of volunteers in certain experimental protocols generated less conclusive and decisive results in terms of statistical power”.

29) “Concerning the devices’ position during activity monitoring (such as walking and

turning), most of the papers agreed that a site close to the center of mass such as the waist [29, 322 30, 46] and lower back [26, 31, 32, 35-37, 41, 43, 44] was ideal for impaired gait analysis.” Why? This needs to be biomechanically justified, since it is relevant for clinical implementation of the devices.

Please see the answer in the comment 30.

30) “consequently, a sensor attached on the limbs would capture recordings with large variations in gait patterns during activity monitoring.” This a wrong statement in the current state.

We have addressed comments 29 and 30 by justifying clinically the choice of the authors of adopting a single sensor close to the center of mass (lines 335-342). 

“Devices’ number and placement were various, depending on the outcomes measured. Concerning impaired locomotion, the center of mass was extensively used in literature to measure movement performance and level of stability [60–62]. Accordingly, to monitor activities such as walking and turning, most of the papers agreed to adopt a single sensor worn close to the waist [36,37,53] and lower back [33,38,39,42–44,48,50,51]. Besides, PwP may exhibit asymmetric walk due to the different level of impairment of the lower limbs, characterized by a reduction in walking speed, shuffling steps, and limited foot lifting [3]. Consequently, a sensor attached on the single limb would capture recordings with large variations in gait patterns and it would give just a partial overview of the patient’s status”. 

Added refeences:

Fazio P, Granieri G, Casetta I, Cesnik E, Mazzacane S, Caliandro P, Pedrielli F, Granieri E. Gait measures with a triaxial accelerometer among patients with neurological impairment. Neurological Sciences. 2013 Apr 1;34(4):435-40.

Howell D, Osternig L, Chou LS. Monitoring recovery of gait balance control following concussion using an accelerometer. Journal of biomechanics. 2015 Sep 18;48(12):3364-8.

Betker AL, Szturm T, Moussavi Z. Center of mass approximation during walking as a function of trunk and swing leg acceleration. In2006 International Conference of the IEEE Engineering in Medicine and Biology Society 2006 Aug (pp. 3435-3438). IEEE.

31) “manifest differently and on various locations from person to person [2].” due to laterality?

As the reviewer correctly mentioned, the PD condition manifest differently and not always on the same limb. We have changed this statement accordingly (lines 343-346):

“Sensor positioning and number is also crucial for the assessment of multiple symptoms on different subjects. In fact, tremor, dyskinesia, bradykinesia, and other PD related motor fluctuations affect upper and lower limbs differently depending on the manifestation and stage of the disease [3]”.

32) “should be always considered to guarantee the long-term acceptability of the system.” for what purpose? This needs to be developed and justified

We have addressed comments 26 and 32 in the following paragraph (lines 351-359).

“Finally, given the potential continuous long-term adoption of wearable systems by PwP, aspects which were neglected in the identified papers, such as a system’s comfort of use, set-up process, instructions for use, support, aesthetics and display, should always be considered to guarantee long-term acceptability and efficacy of the system. For instance, the FDA-approved Parkinson’s Kinetigraph system (PKG), which provides continuous, objective, ambulatory assessments of PD symptoms, has been proved to show high patient acceptability, with 81% of the users reporting satisfactory outcomes [63]. These considerations are crucial if the final purpose is to gather large datasets and if PwP have to interact on a daily basis with the system”.

Added reference:

Dominey T, Hutchinson L, Pearson E, Murphy F, Bell L, Carroll C. Evaluating the clinical utility of the Parkinson’s KinetiGraph (PKGTM) in the remote management of Parkinson’s disease. Mov Disord. 2018.

Reviewer 2 

1) Why the papers only from 2010 to 2020 were selected for review?

The interest on continuous home monitoring of Parkinson’s disease has been growing steadily during the last decade. With our search strings we were able to identify just 16 and 19 papers in 2011 and 2012 while the number is sensibly higher in 2018 (75 papers) and 2019 (96 papers). We kindly believe that a decade fully captures the state of the art regarding the topic. We also have added the number of articles found for each specific year, kindly refer to comment 14th of the first reviewer. 

2) Figure quality is very poor, difficult to read.

Thank you for underling this issue. We have improved the quality of the figures in the manuscript.

3) Table 2 is not adding much to the manuscript as you already given detail in the section 3.2. You can put it in the supplementary material by highlighting the missing parts instead of just giving them score.

We have removed the table from the manuscript and have added in the supplementary material.

4) Please discuss the specifications of the inertial sensors used in these studies. Also shed some light either these sensors are fit for this purpose, or there is a need of custom-built sensors for passive monitoring of the disease.

The majority of the studies used off-line solutions and off-the-self devices. We believe that the development of custom-built sensors (ad-hoc hardware and on-board algorithms) could improve real-time feedbacks given to clinicians and ultimately to the patients. For this reason, we have modified the paragraph as follows (lines 308-314): 

“The majority of the studies adopted off-the-shelf devices and off-line algorithm solutions. However, a potential implementation of ad-hoc hardware and on-board algorithms could enhance real-time feedbacks and ultimately have a meaningful impact in the life of patients living, for instance, in rural communities and remote areas [35,46]. In both cases, the direct manipulation of raw data, gathered during the free-living acquisitions, avoids the use of aggregated data (i.e. step, distance) generated by “black box” software of commercial devices”.

5) Please revise line 284-288 for better readability

We have modified the paragraph as follows (lines 299-303). 

“Wearable motion sensors are able to monitor PwP outside of standard clinical environments (for example, in private homes or community dwellings), and provide technically and clinically relevant information for clinicians and patients; therefore, a continuous assessment of the pathology may improve the quality of life of PwP, allowing them to preserve their independence and avoid additional disease complications. [12,56,57]”. 

6) Why we need assessment of 7 days? Please discuss its importance.

Thank you for emphasising on this aspect of the assessment. Numerous episodes related with PD are challenging to detect during laboratory-based sessions and long observation windows if multiple days are required due to the complexity of the disease. IMUs provide relevant information (technically and clinically) for clinicians and patients (lines 293-303)

 “…the evaluation of PD requires extensive judgement from highly-trained professionals, yet clinical assessments in a clinical setting provide only a partial overview of the disease’s pathological progression [55]. In addition, numerous episodes related with PD are challenging to detect during laboratory-based short-term observations. To consistently analyse motor symptoms, fluctuations and gait impairments, long observation windows are required due to the complexity and sporadicity of such events [21].

Wearable motion sensors are able to monitor PwP outside of standard clinical environments (for example, in private homes or community dwellings), and provide technically and clinically relevant information for clinicians and patients; therefore, a continuous assessment of the pathology may improve the quality of life of PwP, allowing them to preserve their independence and avoid additional disease complications. [12,56,57]”. 

And also (lines 92-100):

“Former studies, such as the one by Bolem et al. (2001) [19], have reported that PwP walk better when observed rather than when unsupervised in their daily lives. This is a consequence of the well-known “Hawthorne observation effect” [20]: free-living activities involve a combination of tasks with varying complexities, challenges and distractions that may reduce attention. In addition, numerous episodes related with PD are challenging to detect during laboratory-based observation because of their complexity (i.e. the on/off phenomenon) or rarity (i.e. freezing of gait phenomenon) [21]. As a consequence, a thorough evaluation of a PwP requires the data to be gathered during long observation windows while patients go ahead with normal every day activities”. 

7) Please discuss in detail, what kind of features can give better discrimination between PwP and controls in the free-living environment and why? How these features are going to be different from the controlled environment?

In section “Results: data norms, validity and predictive power of the included studies” we underlined the features that give better discrimination between PwP and controls in a free-living environment (lines 229-242):

“Yoneyama et al. (2013/2014) [36,37] found that the average duration of the gait cycle was longer in PwP (1.16 ± 0.20 s) compared to controls (1.08 ± 0.19 s; p < 0.001). Similarly, Del Din et al. (2016) [44] reported that Parkinsonians walked with slower and shorter steps (step velocity:1.038 ± 0.422 m/s and 1.103 ± 0.411 m/s for PD and controls, respectively; p < 0.001). Moreover, PwP presented less consistent (e.g. step time variability: 0.175 ± 0.156 s for control and 0.181 ± 0.179 for PD; p = 0.07) and asymmetric (e.g. step time asymmetry: 0.093 ± 0.086 for control and 0.098 ± 0.142 for PD; p = 0.116) walking patterns [44], with fluctuations in kinematics and frequency measures compared to healthy subjects [31,33,44].

Three studies also investigated turning [38,40,48] and confirmed that PwP take shorter turns (2.0 s and 2.2 s for PD and control, respectively; p = 0.001) with smaller angles (92.0° and 95.2° for PD and control, respectively; p = 0.001) [38]. In addition, PwP completed the turning movement at a slower pace than controls (turn mean velocity: 38 ± 5.7 °/s and 43.3 ± 4.8 °/s, respectively; p = 0.04) and with a greater number of steps (mean number of steps: 3.2 ± 0.8 and 1.7 ± 1.1, respectively; p = 0.04) [40]”.

In addition, in “Discussion” in particular in the section “Aim, outcome measures, type of analyses, and results”. We have debated which parameters can clearly differentiated between the PD and healthy populations (lines 362-375): 

“Kinematic parameters, such as duration of gait cycle, step length, and velocity, were clearly differentiated between the PD and healthy populations. In fact, PwP walked slower and with shorter steps [36,37,44]. Less consistent gait patterns with major fluctuations in kinematics and frequency measures were also observed [31,33,44]. Findings also underlined differences in turning [38,40,48], showing patients taking shorter turns with smaller angles and completing the turning movement slower and with a greater number of steps. Concerning the risk of falling, the relationship between the level of activity and impairments is still a matter of debate among the scientific community. On one side, more active patients could be more susceptible to falls since they are exposed to more unsafe situations, but on the other hand they could be at a lower risk of falling due to a better general health condition. Two reviewed articles estimated that falls occurred significantly more frequently in PwP with a less consistent walking pattern [39,50], while fallers seemed to have a reduced capability to regulate gait due to a partial loss of postural stability [64]. Inertial wearable device can detect such impaired walking patterns and predict future falls even in patients with no previous fall history [39]”.

Concerning the difference between supervised and unsupervised environment, we kindly believe that the main goal of this paper is to provide future directions for scientists that aim to start a continuous assessment of PwP for prolonged time. We kindly decided to not include these differences because not part of the main aim of the literature review”.

8) Please indicate the future directions in bullet form based on the summary of these papers.

Thanks the reviewer for the comment. We kindly believe that bullet points may alter the fluidity and readability of the text. However, we have listed the future directions as follow (lines 416-425):

“In conclusion, future studies commencing an assessment of PwP for prolonged time periods may look into the a) development and testing of dedicated hardware and software for real-time feedback that would also permit the interaction between clinicians and patients, and b) the incorporation of digital versions of diaries with alerts and prompts in the study’s design that would allow the correlation between quantitative measurements and self-reported outcomes. Additionally, characteristics which were ignored by researchers, such as the system’s comfort of use, set-up process, instructions for use, support, aesthetics and display, need to be strongly considered. These reflections are fundamental for the efficacy of a health care system that will be used mostly by older people in a social environment and it should not affect patients physically or psychologically [12,56,57,65–70]”.

---

## [Decision Letter · Decision Letter 1]

21 Jan 2021

Continuous home monitoring of Parkinson’s disease using inertial sensors: a systematic review

PONE-D-20-24431R1

Dear Dr. Sica,

We’re pleased to inform you that your manuscript has been judged scientifically suitable for publication and will be formally accepted for publication once it meets all outstanding technical requirements.

Kind regards,

Fabio A. Barbieri, PhD

Academic Editor

PLOS ONE

Additional Editor Comments (optional):

Reviewers' comments:

Reviewer's Responses to Questions

**Comments to the Author**

1. If the authors have adequately addressed your comments raised in a previous round of review and you feel that this manuscript is now acceptable for publication, you may indicate that here to bypass the “Comments to the Author” section, enter your conflict of interest statement in the “Confidential to Editor” section, and submit your "Accept" recommendation.

Reviewer #1: (No Response)

Reviewer #2: All comments have been addressed

2. Is the manuscript technically sound, and do the data support the conclusions?

Reviewer #1: Yes

Reviewer #2: Yes

3. Has the statistical analysis been performed appropriately and rigorously? 

Reviewer #1: Yes

Reviewer #2: N/A

4. Have the authors made all data underlying the findings in their manuscript fully available?

Reviewer #1: Yes

Reviewer #2: Yes

5. Is the manuscript presented in an intelligible fashion and written in standard English?

Reviewer #1: Yes

Reviewer #2: Yes

6. Review Comments to the Author

Reviewer #1: I appreciate how the authors addressed both reviewer's comments. I believe the manuscript has improved its quality to a level of acceptance.

Thank you for considering our comments.

Reviewer #2: Thanks for addressing all the points.

One missing point, apart from the continuous monitoring of PD in real world, usually 3 consecutive days for reliable estimation of gait characteristics are enough.

7. PLOS authors have the option to publish the peer review history of their article (what does this mean?). If published, this will include your full peer review and any attached files.

Reviewer #1: **Yes: **Encarna M. Micó Amigo

Reviewer #2: **Yes: **Rana Zia Ur Rehman

---

## [Editor Report · Acceptance letter]

25 Jan 2021

PONE-D-20-24431R1 

Continuous home monitoring of Parkinson’s disease using inertial sensors: a systematic review 

Dear Dr. Sica:

I'm pleased to inform you that your manuscript has been deemed suitable for publication in PLOS ONE. Congratulations! Your manuscript is now with our production department. 

Kind regards, 

on behalf of

Dr. Fabio A. Barbieri 

Academic Editor

PLOS ONE